# Judging What We Cannot Solve: A Consequence-Based Approach for Oracle-Free Evaluation of Research-Level Math

**Guijin Son** [1][2]  **Donghun Yang** [3]  **Hitesh Laxmichand Patel** [4]  **Hyunwoo Ko** [2]
**Amit Agarwal** [4]  **Sunghee Ahn** [1]  **Kyong-Ha Lee** [3]  **Youngjae Yu** [1]

## Abstract

Recent progress in reasoning models suggests that generating plausible attempts for research-level mathematics may be within reach, but verification remains a bottleneck, consuming scarce expert time. We hypothesize that a meaningful solution should contain enough method-level information that, when applied to a neighborhood of related questions, it should yield better downstream performance than incorrect solutions. Building on this idea, we propose **Consequence-Based Utility**, an oracle-free evaluator that scores each candidate by testing its value as an in-context exemplar in solving related yet verifiable questions. Our approach is evaluated on an original set of research-level math problems each paired with one expert-written solution and nine LLM-generated solutions. Notably, Consequence-Based Utility consistently outperforms reward models, generative reward models, and LLM judges on ranking quality. Specifically, for GPT-OSS-120B it improves Acc@1 from 67.2 to 76.3 and AUC from 71.4 to 79.6, with similarly large AUC gains on GPT-OSS-20B (69.0 to 79.2). Furthermore, compared to LLM-Judges, it also exhibits a larger solver–evaluator gap, maintaining stronger correct–wrong separation even on instances the underlying solver often fails to solve.

## 1. Introduction

For a mathematical hypothesis to be accepted as scientific knowledge, it must undergo extensive review and validation. Yet many recent efforts to advance science with LLMs (Gottweis et al., 2025) emphasize hypothesis generation (Zhou et al., 2024; Radensky et al., 2024) and experimental plan-

[1]Seoul National University [2]OnelineAI [3]KISTI [4]ORACLE. Correspondence to: Guijin Son <guijin.son@snu.ac.kr>.

*Proceedings of the 43rd International Conference on Machine Learning*, Seoul, South Korea. PMLR 306, 2026. Copyright 2026 by the author(s).

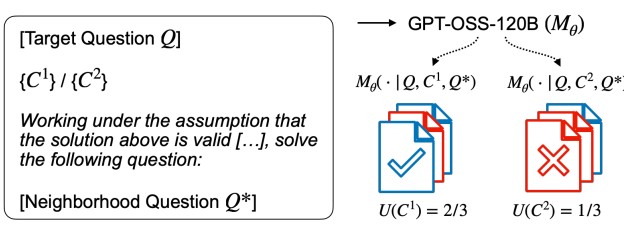

*Figure 1.* **Consequence-Based Utility for solution validation.** We use GPT-OSS-120B as the solver $M_\theta$ and score each candidate solution by its induced accuracy on neighborhood questions $Q^*$; $U(C^1) > U(C^2)$ suggests $C^1$ is more likely correct.

ning (Goel et al., 2025), while giving comparatively less attention to rigorous validation. Accordingly, this step is largely dependent on either human experts (Georgiev et al., 2025), which are costly to scale, or LLM judges (including agentic systems) (Lu et al., 2024; Zhu et al., 2025; Panigrahi et al., 2026), that are often unreliable (Son et al., 2024b; 2025a) and biased (Ye et al., 2024). These limitations motivate the need for better methods for hypothesis validation.

In this work, we introduce **Consequence-Based Utility**, a novel approach to validate a set of candidate solutions without access to ground-truth answers. As shown in Figure 1, we prompt a solver $M_\theta$ with a research-level question $Q$ and $C^{(i)}$ as in-context exemplars. For each $C^{(i)}$, we measure the solver's accuracy on a closely related neighborhood problem $Q^*$ and use the resulting accuracy as its utility score. Intuitively, a candidate that induces higher accuracy on $Q^*$, provides more helpful information for $Q$ and is therefore more likely to be correct. It should be noted that *Consequence-Based Utility is designed to focus on research-level questions*, those remaining open to LLMs today. We therefore focus on genuine research-level questions that remain out of reach for today's LLMs, and curate EXPERTMATH consisting of 192 expert-written problems and 425 LLM-Generated questions. Half of the expert-written questions remain open to leading models (e.g., GPT-5 and Gemini-3-Pro). In this dataset, our method outperforms oracle-free baselines such as reward models, generative reward models, and LLM judges. For instance, as an LLM-Judge, GPT-OSS-120B achieves Acc@1 = 67.21 and AUC

= 71.42; under Consequence-Based Utility, these increase to 76.27 and 79.63, respectively. Moreover, Consequence-Based Utility exhibits a larger solver–evaluator gap than LLM judges, preserving a stronger separation between correct and incorrect solutions even for questions the model fail to solve. This makes it particularly well-suited for evaluating research-level questions. Finally, our error analysis reveals that these gains arise from Consequence-Based Utility more reliably downranking solutions with incorrect reasoning, unjustified compression, or unjustified interpretation, and being less sensitive to stylistic cues and authority-like statements that are known to mislead LLM judges (Ye et al., 2024; Moon et al., 2025).

Our contributions are summarized as follows:

- We propose **Consequence-Based Utility**, an oracle-free method for validating candidate solutions via downstream performance on neighborhood questions.

- We release EXPERTMATH a collection of 192 expert-written research-level math problems with author solutions, along with 425 LLM-generated problems.

- We show that CBU consistently outperforms oracle-free baselines (LLM-judges, reward models, and generative reward models), and identify judge failure modes that CBU reliably penalizes through error analysis.

- We provide a practitioner's guide for CBU, including how to construct neighborhood questions and how many rollouts are needed for stable utility estimates.

## 2. Preliminary and Related Works

### 2.1. Call for Oracle-Free Validation in Math

Recent case studies indicate that LLMs can meaningfully assist professional mathematicians on genuine open or previously unsolved research problems. In late 2025, publicly documented human–LLM collaborations (i) established point convergence of Nesterov's accelerated gradient method (Jang & Ryu, 2025), (ii) produced a finite counterexample to a "majority optimality" conjecture in non-interactive correlation distillation with erasures (Ivanisvili & Xie, 2025), and (iii) determined the sharp minimax-optimal error rate for robust density estimation under Wasserstein-bounded contamination (Dobriban, 2025). Despite the notable progress, however, these reports underscore that current models are high-variance generators rather than reliable autonomous theorem provers: Jang & Ryu (2025) reports that ChatGPT generated *"numerous arguments, approximately 80% of which were incorrect,"* Dobriban (2025) notes that GPT-5 *"glossed over details that sometimes took days of work to fill in,"* and Schmitt (2025) observes that *"Some models claimed false counterexamples."* Consequently, progress still depends on professor-level triage.

Experts must reject hallucinated proof attempts, repair missing steps, and translate ideas into checkable arguments before any result is safe to trust or share. These experiences motivate the need for oracle-free validation: scalable validation mechanisms that can filter and score candidate research outputs without requiring a scarce domain-expert oracle for each attempt.

### 2.2. Existing Oracle-Free Validators.

We model a *candidate solution* as an object $C \in \mathcal{C}$ (e.g., a proof sketch, lemma chain, or an algorithmic construction) for a research question $Q \in \mathcal{Q}$. A generator LLM $M_\theta$ induces a conditional distribution over candidates,

$$C^{(i)} \sim p_\theta(\cdot \mid Q), \qquad i = 1, \dots, N.$$

In an idealized setting, there exists a (typically unavailable) correctness oracle

$$O(Q, C) \in \{0, 1\},$$

which returns $1$ iff $C$ is fully correct (and $0$ otherwise). "Oracle-free validation" replaces $O$ with a *validator $V$* that outputs a score used for selection or ranking:

$$V : \mathcal{Q} \times \mathcal{C} \to \mathbb{R}, \qquad \widehat{C} = \arg\max_{i \in [N]} V(Q, C^{(i)}).$$

Below, we formalize three widely used validators: consistency voting (Wang et al., 2022), reward models (Ouyang et al., 2022), and LLM judges(Zheng et al., 2023).

**(1) Majority (consistency) voting.** Majority voting assumes that each candidate $C$ deterministically induces a discrete prediction $A(C) \in \mathcal{A}$ (e.g., a numeric answer or yes/no). Given $N$ i.i.d. samples $C^{(1:N)}$ with induced answers $A^{(i)} := A(C^{(i)})$, the majority-vote answer is $\widehat{A}_{\mathrm{mv}} := \arg\max_{a \in \mathcal{A}} \sum_{i=1}^{N} \mathbf{1}\{A^{(i)} = a\}$. This approach may be effective when correctness is tightly tied to a single discrete final answer, as in contest-style or short-answer math. For research problems, however, the validity of a solution often cannot be reduced to a discrete label. We therefore exclude majority voting from our study.

**(2) Reward models.** A reward model is a scoring function that approximates solution "quality" in a cardinal way:

$$R_\phi : \mathcal{Q} \times \mathcal{C} \to \mathbb{R}, \qquad V_R(Q, C) = R_\phi(Q, C).$$

In use, an RM provides a scalar signal for ranking and optimization. A common training approach fits $R_\phi$ from pairwise preferences using a Bradley–Terry model (Yuan et al., 2024; Hong et al., 2025): for a comparison $(Q, C_a, C_b)$, the probability that $C_a$ is preferred is

$$p_\phi(C_a \succ C_b \mid Q) = \sigma\big(R_\phi(Q, C_a) - R_\phi(Q, C_b)\big).$$

Parameters $\phi$ are then learned by maximum likelihood (i.e., a standard logistic preference loss). To scale RMs at inference time, process reward models (PRMs) (Zhang et al., 2025b) and generative reward models (GenRMs) have been proposed. In our setting, we default to **GenRMs** (Zhang et al., 2024), as recent work suggests PRMs can be less stable than outcome-level scoring (Guo et al., 2025; Son et al., 2025b), and current practice increasingly emphasizes generative evaluators (Blakeman et al., 2025; Liu et al., 2025b). A GenRM produces an evaluation string $Z \in \mathcal{Z}$ (typically a short critique containing an explicit numeric score),

$$Z \sim p_\phi(\cdot \mid Q, C),$$

and a deterministic parser $\mathsf{score} : \mathcal{Z} \to \mathbb{R}$ extracts a scalar reward. This induces a single-sample score,

$$R_\phi^{\text{gen}}(Q, C) = \mathsf{score}(Z).$$

**(3) LLM judges.** An LLM judge is a model $J_\psi$ that we prompt to evaluate a candidate solution $C^{(i)}$. In common practice, the judge first produces a natural-language critique $Z^{(i)}$ and then outputs a discrete rating $Y^{(i)}$. In this paper, the rating is an integer score on a 1–10 scale,

$$(Z^{(i)}, Y^{(i)}) = J_\psi(Q, C^{(i)}), \qquad Y^{(i)} \in \mathcal{Y} = \{1, \dots, 10\}.$$

We reduce the judge output to a numeric validator by taking the score directly,

$$V_J(Q, C^{(i)}) = s\left(Y^{(i)}\right), \qquad s(y) = y.$$

## 3. Consequence-Based Utility

**Motivation and hypothesis: utility via "support by consequences."** When a target question $Q$ is difficult to verify directly (e.g., because a reference answer is unavailable or costly to obtain, or because the solution is long and subtle), a widely adopted method in mathematics is the "support by consequences" perspective: rather than scoring the claim in isolation, we assess it by the breadth and coherence of what it enables. A canonical example is the Riemann Hypothesis, which remains unproven yet underwrites many sharp conditional results across analytic and algorithmic number theory (e.g., Von Koch (1901); Rosser & Schoenfeld (1975); Miller (1975); Bach (1990)). Analogously, we treat each candidate solution $C^{(i)}$ as a provisional *hypothesis* about $Q$ and evaluate its quality by transfer: even when $C^{(i)}$ cannot be validated reliably on $Q$ itself, it may still be judged by how consistently it provides useful guidance for solving related, verifiable questions in a neighborhood around $Q$.

Our hypothesis is therefore: *correct (or near-correct) candidates contain method-level information that transfers to a neighborhood of related questions and yields consistently higher downstream performance, and vice-versa.*

**Implementation in the LLM setting.** Given a problem $Q$, we sample $N$ candidate solutions $C^{(i)}$ from the generator $M_\theta$. Because the ground-truth oracle $O(Q, C)$ is unavailable, we estimate a candidate's usefulness by measuring how well it transfers to a neighborhood of related problems for which correctness is verifiable (e.g., previously solved or otherwise easier instances). We define this set of neighborhood questions as $\mathcal{N}(Q)$. For a fixed candidate $C$, we condition $M_\theta$ on $(Q, C)$ and ask it to solve each $Q^* \in \mathcal{N}(Q)$. We score each rollout using a verifier $v(Q^*, \tilde{C}) \in \{0, 1\}$ that checks whether the completion $\tilde{C}$ constitutes a correct solution for $Q^*$ under our pipeline. We define the Consequence-Based Utility as the average accuracy on these variants:

$$U(C) = \frac{1}{|\mathcal{N}(Q)|} \sum_{Q^* \in \mathcal{N}(Q)} \mathbb{E}_{\tilde{C} \sim M_\theta(\cdot | Q, C, Q^*)} \left[ v(Q^*, \tilde{C}) \right].$$

In practice, we estimate this by sampling $T$ independent rollouts $\tilde{C}_t \sim M_\theta(\cdot \mid Q, C, Q^*)$ for each $Q^*$ and averaging their scores:

$$\widehat{U}(C) = \frac{1}{|\mathcal{N}(Q)| \, T} \sum_{Q^* \in \mathcal{N}(Q)} \sum_{t=1}^{T} v\left(Q^*, \tilde{C}_t\right).$$

**In-context learnability as a correctness signal.** Prior work have leveraged in-context performance as a proxy to value examples and demonstrations (Chang & Jia, 2023; Nguyen & Wong, 2023; Xie et al., 2024). Relatedly, context conditioning also serves as a training signal, e.g., by distilling from a teacher that observes privileged traces while the student observes only the question (Zhao et al., 2026). Despite this progress, in-context valuation is used mainly for data curation, retrieval, attribution, or training, with limited use as an oracle-free *verification* mechanism. Our work differentiates from past efforts by leveraging in-context learnability to validate candidate solutions by measuring their downstream consequences on neighborhood problems.

## 4. Experiment Setup

### 4.1. Collecting Research-Level Math Problems

We start from 70 faculty-authored, hand-crafted questions, spanning three broad areas and including keywords such as, but not limited to, **representation theory and algebraic combinatorics** (Hecke algebra, universal Coxeter system, Kazhdan–Lusztig polynomials, Polo's algorithm, Brenti's conjecture), **geometry (algebraic and differential)** (Kollár–Johnson threefold, $\mathbb{Q}$-Fano, Ricci lower bounds), and **homotopy theory and homotopical methods** (homotopical algebra, $p$-adic homotopy theory, Shafarevich extensions).

Table 1 highlights the challenging nature of our dataset, EXPERTMATH , by comparing it among established math

**(A) Target Question $Q$**

Let $(W, S)$ be the Coxeter system of type $D_8$. Consider the asymptotic Hecke algebra $J = \oplus_{w \in W} \mathbb{Z} t_x$ of $(W, S)$ in the equal parameter case, as defined by Lusztig. Let $\mathscr{D} \subseteq W$ be the set of distinguished involutions.

If $(\sum_{w \in \mathscr{D}} t_w)^2 = \sum_{x \in W} a_x t_x$, compute the number $\sum_{x \in W} a_x$.

**(B) Candidata Solutions** $(C^{1:N})$

**C1 (Expert-written, correct)**
Unit property in $J$ gives $\sum_x a_x = \#\mathscr{D}$; $\mathscr{D} \leftrightarrow$ left KL cells. Type $D_8$: # left KL cells =11504.

**C2 (LLM-written, correct)**
$\sum_{d \in \mathscr{D}} t_d$ is the identity in $J$, so $(\sum_{d \in \mathscr{D}} t_d)^2 = \sum_{d \in \mathscr{D}} t_d)$. Hence $a_x = 1$ iff $x \in \mathscr{D}$,

so $\sum_x a_x = |\mathscr{D}| = \#.$ {left KL cells} Type $D_8$: $\sum_x a_x$=11504.

**C3 (LLM-written, wrong**
Correctly identifies $\sum_x a_x = |\mathscr{D}|$, but incorrectly equates the number of left KL cells

with the number of irreducible representations, yielding the wrong value 170.

**(C) Neighborhood Question $Q*$**

**Q1** Let $W_1$ and $W_2$ be the Coxeter groups of types $E_7$ and $B_7$, respectively. Compute the number $\#\{x \in W_1 \mid a(x) = 1\} + \#\{x \in W_2 \mid a(x) = 1\}$, where $a$ denotes Lusztig's $a$-function.

**Q2** Let $(W, S)$ be the Coxeter system of type $E_7$. Consider the asymptotic Hecke algebra $J = \oplus_{w \in W} \mathbb{Z} t_x$ of $(W, S)$ in the equal parameter case, as defined by Lusztig. Let $\mathscr{D} \subseteq W$ be the set of distinguished involutions. If $(\sum_{w \in \mathscr{D}} t_w)^2 = \sum_{x \in W} a_x t_x$, compute the number $\sum_{x \in W} a_x$.

*Figure 2.* **Example of a target question, candidate solutions, and neighborhood questions from EXPERTMATH.** (A) A target research-level problem on the asymptotic Hecke algebra $J$ of the Coxeter group of type $D_8$. (B) A fixed candidate pool $C^{1:3}$ illustrating three typical solution types appearing in our dataset: an expert-written correct solution $C^1$; an LLM-generated solution that is mathematically correct $C^2$; and a plausible but incorrect LLM-generated solution $C^3$ that makes a subtle conceptual error by conflating the number of left Kazhdan–Lusztig cells with the number of irreducible representations. (C) Two neighborhood questions $Q^*$ derived from $Q$ by modifying the Coxeter type or the associated invariant.

*Table 1.* **Scores indicate EXPERTMATH is substantially harder than AIME 25 and IMProofBench, and comparable to FrontierMath (T1–3).** EXPERTMATH uses Avg@8 for all models except GPT-OSS-120B, which uses Avg@64. AIME 25 uses Avg@10. For IMProofBench, we report the subquestion score, where subquestions are specific, automatically-verifiable components of larger problems; the overall aggregation metric is not specified in the source. FrontierMath (T1-3) uses Avg@8. [1]

| Model | (Ours) | AIME 25 | IMProofB. | F.M. |
|---|---|---|---|---|
| Public | △ | O | X | X |
| # Unsolved | 38 | 0 | 5 | - |
| Gemini-3-Pro | 47.14 | 95.7 | 71.8 | 37.6 |
| GPT-5 | 35.71 | 94.3 | 54.5 | 32.4 |
| Claude-Opus-4.5 | 7.14 | 91.3 | - | 20.7 |
| Claude-Opus-4.1 | - | 80.3 | 38.7 | - |

evaluations. Along with AIME 2025 (MAA), an invitational competition to USAMO, IMProofBench (Schmitt et al., 2025) targets research-level mathematical proof writing, and FrontierMath (Glazer et al., 2024) is explicitly designed as a collection of unpublished, expert-authored problems. The score on EXPERTMATH (7.14–47.14; mean 25.5) indicates higher difficulty than competition-style benchmarks such as AIME 25 (80.3–95.7; mean 91.0), and lower performance than IMProofBench (37.6–71.8; mean 50.7). The absolute scale on our benchmark is closest to FrontierMath (T1–3) (20.7–37.6; mean 30.2). Finally, over half of the collected questions are unsolved by any of the tested models, remaining open to frontier models such as GPT-5 (Singh

et al., 2025) and Gemini-3-Pro (Team et al., 2025).

## 4.2. Neighborhood Questions, Ground Truths, and Candidate Solutions

For each problem, we additionally collect a set of *neighborhood questions*. These questions are author-created variants that preserve the core mathematical idea while perturbing the statement. Authors are instructed to design variants that become straightforward once the original problem is understood (e.g., by reusing the same key lemma or reduction), and to make them slightly easier than the original whenever feasible. In practice, having too many variants tends to become redundant. Accordingly, we cap collection at two variants per original problem. Authors receive approximately \$600 per problem package, which includes the main problem, neighborhood questions, and reference solutions. To the best of our knowledge, EXPERTMATH is the only benchmark at this difficulty that provides expert-written solutions. See Appendix D for further example and details.

Every original problem and neighborhood variant is accompanied by an author-written ground-truth solution. Expert-written solutions range from detailed, multi-page expositions to concise sketches, intuition-driven arguments, or pointers to external results sufficient to reconstruct a full proof. For the ease of automated verification, we require that

---

[1] - (hyphen) denotes an unavailable value, typically because the benchmark is private and organizers did not release the score. Sources: AIME 25 (Artificial Analysis), IMProofBench (`improofbench.math.ethz.ch`), FrontierMath (`epoch.ai/frontiermath`).

*Table 2.* **Validator performance on ranking LLM solutions.** Consequence-Based Utility shows the highest performance across all metrics. Best models are highlighted in **bold**, second best is underlined.

| Models | HumanWin | MeanWin | Acc@1 | Recall@5 | AUC |
|---|---|---|---|---|---|
| **(Generative) Reward Models** | | | | | |
| Qwen3-235B-GenRM | 27.05 | 77.05 | 65.37 | 71.72 | 67.85 |
| Llama3.3-Nemotron-49B-GenRM | 25.71 | 31.43 | 43.47 | 55.36 | 49.57 |
| Qwen2.5-Math-RM-72B | 1.63 | 27.87 | 36.89 | 40.98 | 34.05 |
| AceMath-72B-RM | 0.00 | 12.86 | 8.20 | 29.85 | 20.75 |
| **LLM-Judges** | | | | | |
| Qwen3-235B-A22B | 67.14 | 85.71 | 62.59 | 80.02 | 69.48 |
| Qwen3-30B-A3B | 47.14 | 75.71 | 61.30 | 72.40 | 65.81 |
| GPT-OSS-120B | 48.57 | 81.43 | 67.21 | 76.91 | 71.42 |
| GPT-OSS-20B | 52.86 | 82.86 | 72.13 | 72.06 | 69.03 |
| **Consequence-Based Utility** | | | | | |
| Qwen3-235B-A22B | 81.43 | **90.00** | 73.42 | 74.15 | 71.38 |
| Qwen3-30B-A3B | **85.71** | **90.00** | 75.79 | 78.37 | 76.24 |
| GPT-OSS-120B | 82.86 | **90.00** | **76.27** | **83.04** | **79.63** |
| GPT-OSS-20B | 74.29 | 75.71 | 74.59 | 82.46 | 79.18 |

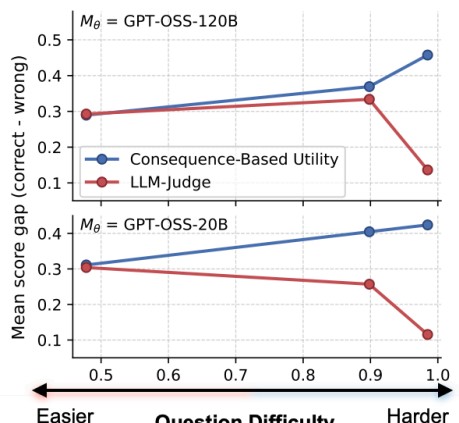

*Figure 3.* **Mean score gap (correct - wrong) versus question difficulty for LLM-Judge and Consequence-Based Utility.**

the final answer be presented in a compact, verifiable form, even when the accompanying writeup is informal. Finally, we construct a pool of LLM-generated candidate solutions for each original question by sampling across a diverse set of models: GPT-OSS-120B, GPT-5, GPT-5 Pro, Gemini-3-Pro, and Gemini DeepThink. We curate nine candidate model solutions, four correct and five incorrect, per problem.[2] Each candidate is manually reviewed in two steps: (i) verifying agreement with the ground-truth final answer, and (ii) reading the derivation to confirm mathematical validity. **The final dataset consists of 192 original research-level math problems (70 original and 122 variants)**, each paired with expert-written solutions and 630 LLM-generated solutions with human validation. See Figure 2 for an example ternary.

### 4.3. Baselines

Given a fixed candidate pool $\{C^{(i)}\}_{i=1}^N$ for each target problem $Q$, we compare Consequence-Based Utility against three standard oracle-free selection baselines: (i) LLM judges, (ii) RMs, and (iii) GenRMs. We use four models, GPT-OSS-20B/120B (Agarwal et al., 2025), and Qwen3-30B-A3B/235B-A22B (Yang et al., 2025) to attempt neighborhood questions conditioned on $(Q, C)$. The same models are used for the LLM-Judges as well. For RM baselines, we use AceMath-RM-72B (Liu et al., 2025a) and Qwen2.5-Math-RM-72B (Yang et al., 2024), two math-specialized reward models. For GenRM baselines, we use Qwen3-Nemotron-235B-A22B-GenRM (Blakeman et al., 2025) and Llama-3.3-Nemotron-Super-49B-GenRM (Wang et al., 2025). The standard template for the two models expects two responses and outputs both per-response and pairwise

signals. In our experiments, we provide the candidate as the first response and a fixed dummy string as the second, and parse only the per-response helpfulness score. Excluding the deterministic RM for which we run a single scoring pass, GenRMs and LLM-Judges are repeated 64 times independently. This is to match its inference cost with Consequence-Based Utility. Across all settings, models are allowed to reason up to 16k tokens, with the temperature set to the recommended value. Since released reward models typically have much shorter native context windows, we apply RoPE scaling (Chen et al., 2023) to support longer inference. See Appendix E for prompts used in our evaluations.

### 4.4. Evaluation Metrics

Each baseline outputs a single scalar score per candidate solution. Since our dataset provides binary labels rather than graded quality, we do not evaluate score calibration. Instead, we measure whether scores rank and separate correct solutions above incorrect ones. We report five higher-is-better metrics: **Acc@1** (whether top-ranked is correct), **Recall@5** (the fraction of correct solutions recovered in the top five), **AUC** (pairwise separability between correct and wrong solutions, with ties partially credited), **HumanWin** (likelihood of human-written solution scores above the average wrong solution), and **MeanWin** (likelihood of mean correct score above the average wrong score). When multiple variants of the same original question are available, we average over variants. See Table 6 for formal definitions.

## 5. Main Results

**Consequence-Based Utility (CBU) outperforms all baselines.** Table 2 shows a clear hierarchy among the evaluated methods. Reward model baselines perform worst (e.g.,

---

[2]GPT-5 Pro and Gemini DeepThink were added with tool use (web search and code execution) to increase solution diversity.

AceMath-72B-RM attains 20.75 AUC), which is expected given their much smaller compute budget (1/64 of the roll-outs used by other methods) (Lee et al., 2025a). LLM judges are substantially stronger, but Consequence-Based Utility consistently improves over LLM-judge scoring when using the same backbone. For example, with Qwen3-235B-A22B, CBU achieves 71.38 AUC, exceeding both the corresponding LLM judge (69.48) and Qwen3-235B-GenRM (67.85). For GPT-OSS-120B, switching from LLM-judge scoring to CBU improves every metric, with gains ranging from +6.13 on Recall@5 (76.91 to 83.04) to +34.29 on HumanWin (48.57 to 82.86). Similar improvements hold for Qwen3-30B-A3B and GPT-OSS-20B. The main exception is Qwen3-235B-A22B on Recall@5, where the LLM judge outperforms by 5.87 points (80.02 vs. 74.15). Consistent with Figure 7, this appears to stem from overconfident scoring that increases top-5 hit rate while weakening fine-grained ranking. Notably, CBU yields especially large gains on HumanWin even when MeanWin is already high, suggesting better alignment with expert evaluation. We attribute this to a stylistic mismatch: human-written solutions are often terse and intuition-driven, whereas LLM judges can overweight surface cues such as verbosity and canonical formatting (Saito et al., 2023; Ye et al., 2024); CBU is less sensitive to these presentation features.

**Consequence-Based Utility is better in evaluating candidates for questions they cannot solve.** Solve-to-Judge gap (Sun et al., 2025) denotes the disparity between a model's ability to judge a solution and its ability to solve the underlying problem. Figure 3 plots the mean score gap between correct and incorrect solutions versus question difficulty, measured by $1 - \text{avg@64}$ (0 = fully solved; 1 = essentially unsolved). Even in the hardest regime ($1 - \text{avg@64} \approx 1$), both LLM-Judge and CBU exhibit nonzero separation, consistent with concurrent findings that models can distinguish correct from incorrect solutions on instances they cannot solve themselves (Nie et al., 2025). As difficulty increases, however, the evaluators diverge. The judge's separability drops sharply, whereas CBU remains robust, making it better suited for the high-difficulty tail characteristic of research-level problems. This pattern is expected in part because CBU uses neighborhood performance as a proxy for correctness, which becomes less informative on easy instances where the solver succeeds regardless of conditioning (e.g., it solves without help, or repairs errors from an incorrect candidate). More broadly, the two methods reflect different evaluation modes. **LLM-Judges resemble a code review**: they inspect a single reasoning trace for plausibility and consistency, which becomes unreliable when incorrect solutions appear superficially coherent and errors are subtle. In contrast, **CBU resembles a unit test**: it scores a candidate by its downstream consequences, whether conditioning on it improves performance

*Table 3.* Predictive performance of score-based feature sets across models. For each backbone (GPT-OSS-20B, GPT-OSS-120B, Qwen3-30B-A3B, Qwen3-235B-A22B), we train a logistic regression binary classifier to predict the label using three alternative feature configurations: GenRM (G), LLM-Judge (J), and Consequence-Based Utility (U).

| Method | G-20B | G-120B | Q3-30B | Q3-235B |
|--------|-------|--------|--------|---------|
| (G) | - | - | - | 54.61 |
| (J) | 63.05 | 64.66 | 58.06 | 66.67 |
| (U) | 73.09 | _73.49_ | 76.31 | 72.69 |
| (J) + (U) | _73.90_ | 73.90 | _76.61_ | **79.65** |

on neighborhood questions, providing a signal that remains informative when direct inspection becomes harder.

**Consequence-Based Utility scores are more predictive of correctness.** Table 3 evaluates how well each validator's scalar score predicts binary correctness by fitting a logistic-regression classifier per backbone and reporting accuracy. Across all four backbones, training on the Consequence-Based Utility score (U) outperforms training on the LLM-judge score (J), with gains ranging from 6.02 points (Qwen3-235B-A22B) to 18.25 points (Qwen3-30B-A3B). This indicates that (U) provides a more linearly separable signal of correctness than (J). Moreover, using both scores together further improves accuracy (e.g., GPT-OSS-20B: 73.09 to 73.90; Qwen3-235B-A22B: 72.79 to 79.65), suggesting that Consequence-Based Utility and LLM-Judges capture complementary information.

# 6. Additional Analysis

Earlier, we showed that Consequence-Based Utility outperforms standard oracle-free validations. In this section, we investigate why this advantage arises and report empirical observations that help explain the performance gap.

**Consequence-Based Utility reduces overconfidence on wrong solutions and better preserves human-written correctness signals.** Figure 5 reports, for each solution type, the probability that a validator assigns an above-average score, $\Pr[s(C) - \bar{s} > 0]$, where $s(C)$ is the validator's score for a candidate and $\bar{s}$ is the validator's mean score over the candidate set for the same instance. Across all models, LLM-judges are more likely than CBU to score LLM-written correct solutions above the mean across all backbones (e.g., Qwen3-235B-A22B shows 0.90 vs. 0.52). In contrast, for human-written correct solutions, the trend reverses. CBU assigns above-mean scores more often than the judge (e.g., GPT-OSS-120B: 0.57 vs. 0.44, and Qwen3-30B-A3B: 0.57 vs. 0.46). Another discrepancy appears on incorrect solutions. LLM-judges are more likely to score wrong answers above the mean, and for Qwen3-30B-A3B and Qwen3-235B-A22B more than half of wrong solutions

| **Incorrect Reasoning** | *"Any tree on $\{1, \ldots, n\}$ can be formed from a tree on $\{1, \ldots, n-1\}$ by adding the vertex $n$ in one of two ways:"* |
|---|---|
| *(Explanation)* Those two operations force $\deg(n) \in 1,2$, but many labeled trees have $\deg(n) \geq 3$. A concrete counterexample (for $n = 4$) is the star centered at 4 with edges $(4,1), (4,2), (4,3)$, where $\deg(4) = 3$. | |
| **Unjustified Compression** | *"Thus, the distribution is symmetric: $a = d$, $b = c$, and $a + d = 60$. Assume $a = 1$, $b = 59$, $c = 50$, $d = 1$.* |
| *(Explanation)* The earlier equations only imply relations like $a = d$, $b = c$, and $a + d = 60$, the logic behind choosing $(1,59,59,1)$ is not introduced. | |
| **Unjustified Interpretation** | *"The number of outgoing edges from an element $w$ is equal to the number of simple reflections $s$ such that $w < ws$ in the Bruhat order".* |
| *(Explanation)* This assumes "strong Bruhat graph" edges come only from simple reflections, which is not justified and may conflict with the standard meaning (strong Bruhat typically involves reflections/covers), changing the distribution. | |
| **External References** | *"It is a known result that for any integer $d \geq 1$, the stretched Littlewood-Richardson coefficient $c^{d\lambda'}_{d\mu', d\nu'}$ is given by a polynomial in $d$, say $P(d)$."* |
| *(Explanation)* The solution omit explanation for a foundational step without stating the precise theorem. Since later steps depend on properties of $P(d)$, this is a reliance on authority rather than demonstrated reasoning within the solution. | |

*Figure 4.* **Illustrative excerpts from incorrect solutions of each error category.** Each row shows a representative quoted snippet (top) and a brief explanation of why it is incorrect or insufficient (bottom). We use four non-exclusive labels: incorrect reasoning, unjustified compression, unjustified interpretation, and external references.

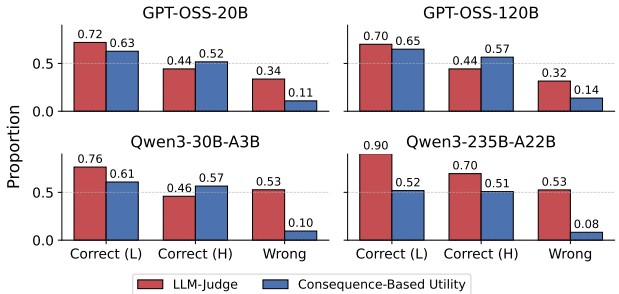

*Figure 5.* **Above-average scoring probability by solution type and backbone.** Each bar measures, $\Pr[s(C) - \bar{s} > 0]$, or how likely a validator is to score a solution above its own typical score on that question, shown separately for LLM-written correct solutions (Correct (L)), human-written correct solutions (Correct (H)), and incorrect solutions (Wrong).

exceed the mean (both 0.53). CBU largely avoids this failure mode, with only 0.08–0.14 of wrong solutions scoring above the mean. Taken together, the performance gap between CBU and LLM-judges likely arises from two factors. CBU better recognizes human-written correct solutions and more reliably penalizes incorrect ones.

**Consequence-Based Utility improves validation by penalizing non-reconstructable reasoning.** To understand why CBU outperforms LLM-judges, we conduct a qualitative error analysis by inspecting 112 incorrect question-solution pairs where GPT-OSS-120B assigns a below-mean CBU score but an above-mean LLM-judge score. We leverage GPT-5-Pro to provide initial labels, which are then confirmed by a mathematics PhD student. We annotate four non-exclusive error types: (i) **incorrect reasoning** (invalid steps, contradictions, or wrong calculations), (ii) **unjustified compression** (missing intermediate steps that prevent local reconstruction or transfer), (iii) **unjustified interpretation** (an unstated choice among plausible readings of the statement), and (iv) **external references** (key claims justi-

fied mainly by citing a named result without derivation or conditions).

These cases concentrate in two failure modes. Unjustified compression occurs in 80/112 (71.4%) and incorrect reasoning in 77/112 (68.8%), suggesting that many wrong solutions appear valid to LLM-Judges, especially when they present polished high-level arguments while omitting verification-critical steps. External references are also common (35/112; 31.3%), consistent with evidence that LLM-judges can be influenced by authority-like cues (Jeong et al., 2025; Moon et al., 2025). A plausible explanation on why CBU likely downranks these solutions may be that wrong or underspecified candidates provide little transferable information for solving neighborhood variants, yielding low utility. Overall, we speculate that CBU gains largely come from downranking convincing-looking solutions that lack reconstructable, transferable reasoning.

## 7. A Practitioner's Guide to Consequence-Based Utility

### 7.1. How Many Rollouts to Generate.

By construction, Consequence-Based Utility requires multiple rollouts as it estimates the candidate's correctness by downstream performance. In contrast, an LLM judge can assign a score in a single pass. To ensure that performance gains do not arise from a larger inference budget, we use 64 rollouts for both LLM-Judge and CBU throughout the paper. The two methods also consume comparable numbers of tokens on average (Table 5), so neither enjoys a systematic budget advantage. A natural question is therefore whether 64 rollouts are necessary to estimate CBU reliably.

Figure 6 reports the mean deviation between an $n$-rollout estimate and the 64-rollout reference. For each $n \in 4, 8, 16, 32, 64$, we uniformly subsample $n$ rollouts from

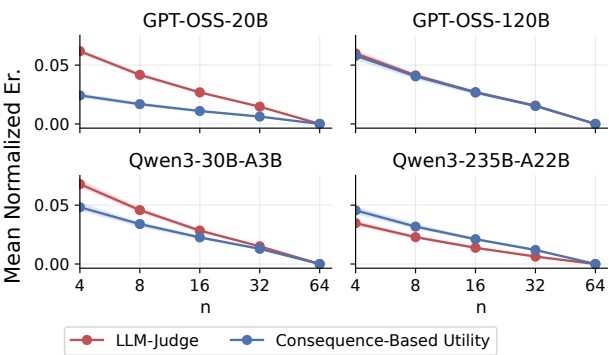

*Figure 6.* **Mean range-normalized absolute error to the 64-rollout reference using** $n \in 4, 8, 16, 32, 64$ **sampled rollouts.** Resampled 200 times using bootstrapping for statistical significance. Normalization uses $[L, U] = [0, 1]$ for CBU and $[0, 10]$ for LLM-judge scores.

the 64-rollout pool, repeat this procedure with 200 bootstrap resamples, and compute the range-normalized absolute error $\frac{|\hat{M} - M|}{U - L}$, where $[L, U] = [0, 1]$ for utility and $[0, 10]$ for LLM-Judge scores. Error decreases monotonically with $n$. CBU converges at a similar rate to LLM-Judges and often faster (notably for GPT-OSS-20B and Qwen3-30B-A3B), while GPT-OSS-120B is nearly identical. Across all backbones, $n \geq 8$ **keeps the mean normalized error below 0.05, indicating that a small number of rollouts already captures most of the signal**.

### 7.2. How to Make Neighborhood Questions.

In our experiments, we use faculty-written neighborhood questions. In practice, however, obtaining expert variants with verified answers can be nearly as difficult as collecting ground truth itself. We therefore study practical alternatives for acquiring $Q^*$ of similar quality. We start from RealMath (Zhang et al., 2025a), which automatically generates graduate-level problems by transforming theorems in mathematics papers. To ensure the questions are sufficiently challenging, we run GPT-OSS-120B for 1024 attempts and retain only instances with intermediate solvability, $0.05 < \text{Avg@}1024 < 0.5$. We then construct neighborhood questions using two approaches. First, we follow explicit "related work" pointers to earlier papers and apply the RealMath transformation to the cited work (e.g., Ortega & Eballe (2022) points to Ortega & Eballe (2021)). Second, we prompt Gemini-3-Pro to generate a closely related variant. We then obtain provisional answers by solving with Gemini-3-Pro, GPT-5-Pro, and Grok-4, and keep only instances where all three agree on the final answer. All candidate solutions are LLM-generated and classified by an LLM-Judge. Because these labels come from model agreement rather than expert verification, this dataset is not suitable for establishing CBU in isolation. Instead, after validating CBU on our expert-written subset, we use it to

*Table 4.* **Performance of GPT-OSS-20B as LLM-Judge and CBU across three datasets.** Each cell reports LLM-JUDGE / CBU scores, the better value is underlined. The two RealMath columns correspond to the two neighborhood-construction procedures described in the text.

| Metric | Daft-Math | Real-Math (1) | Real-Math (2) |
|---|---|---|---|
| Acc@1 | 93.51 / 85.58 | 42.79 / 79.22 | 44.96 / 72.13 |
| AUC | 69.14 / 63.98 | 51.29 / 62.03 | 48.76 / 69.83 |
| MeanWin | 76.62 / 59.74 | 40.26 / 64.94 | 44.87 / 80.33 |

illustrate viable alternatives. Finally, we also consider Daft-Math (Trang, 2025), a collection of contest-level problems paired with lightly transformed variants to have integer answers. The two RealMath subsets and Daft-Math contain 127, 298, and 77 questions, respectively.

Table 4 reports GPT-OSS-20B performance across three datasets. On both RealMath variants, CBU substantially outperforms LLM-judge scoring. In contrast, on Daft-Math, LLM-judge scoring is stronger (e.g., Acc@1 93.51 vs. 85.58). This contrast aligns with our earlier observations on how CBU shows better peformance at questions of higher difficulty. Despite Daft-Math being very close variants (almost identical at core) they are competition level being way easier than the graduate level questions of real-math, so the solver more often succeeds regardless of the in-context exemplar, reducing the discriminative value of utility. Overall, these results suggest that CBU does not require faculty-authored neighborhood questions. LLM-generated neighborhoods can be sufficient when the target questions are challenging for the solver.

## 8. Discussions and Future Work

In this paper, we propose **Consequence-Based Utility**, an oracle-free method that estimates solution correctness from downstream performance when ground truth is unavailable. Across research-level mathematics, CBU consistently outperforms LLM-judges and reward models, and remains effective with both expert-written and LLM-generated neighborhoods. A key limitation may be applicability. Unlike LLM-judges, which exhibit systematic biases but are broadly applicable (Salinas et al., 2025; Son et al., 2024a; He et al., 2025), CBU requires additional effort to construct neighborhood questions. While we show that automated generation is viable (Section 7), reliability depends on the generator's ability to produce sound variants without human oversight. CBU is also most informative when neighborhood difficulty lies in a sweet spot. If $Q^*$ is too easy, the solver succeeds regardless of conditioning, and if too hard, it fails regardless, making neighborhood construction partly model-dependent. Consequently, CBU is best suited to high-stakes settings that demand high-confidence validation for fixed, difficult problems. Future work includes improving

fully automated neighborhood generation, extending CBU beyond mathematics to other STEM domains, and evaluating its effectiveness on genuinely open problems, where both neighborhood construction and correctness assessment are inherently more difficult.

## Acknowledgements

This work was partly supported by the National Research Foundation of Korea(NRF) grant funded by the Korea government(MSIT) (No. RS-2024-00354218), the Technology Innovation Program(RS-2025-25456760, Development of a humanoid robot specialized in chemical processes based on AI foundation model) funded By the Ministry of Trade, Industry and Resources(MOTIR, Korea), Information & communications Technology Planning & Evaluation (IITP) grant funded by the Korea government(MSIT) (No. RS-2026-25522885, Development of a World Foundation Model for Training and Development of Physical AI Systems, No. RS-2021-II211343, Artificial Intelligence Graduate School Program (Seoul National University)), Korea Institute of Science and Technology Information (KISTI) in 2026 (No.(KISTI)K26L3M1C1), aimed at developing KONI (KISTI Open Neural Intelligence), a large language model specialized in science and technology. We express special thanks to the KAIT GPU project. The ICT at Seoul National University provides research facilities for this study.

## Impact Statement

This work aims to make AI-generated mathematical solutions easier to evaluate when full expert verification is difficult. The main positive impact is that it can help experts quickly identify more promising solutions and reduce the time spent reviewing clearly flawed ones. However, our method is not a replacement for human mathematical judgment. Its reliability depends on the quality of the related neighborhood questions used for evaluation. If those questions are poorly designed, the resulting scores may be misleading. Therefore, Consequence-Based Utility should be used as a support tool for expert review, not as an automatic proof verifier.

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

# A. Additional Analyis

## A.1. Output Score Distribution of LLM-Judges

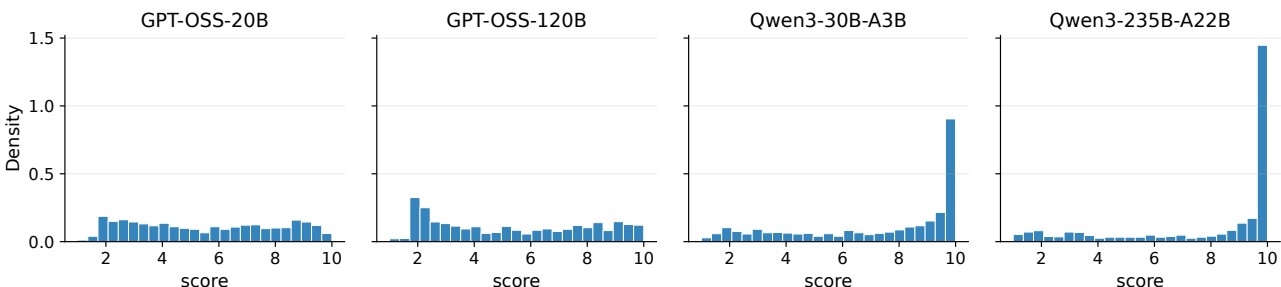

*Figure 7.* **Output score distributions of LLM-judges.** Histograms (density) of judge scores on a 1–10 scale for each backbone over all candidate solutions. GPT-OSS judges spread scores across the range, whereas Qwen judges concentrate near 10, indicating a ceiling effect.

Figure 7 shows the distribution of scalar scores produced by each LLM-judge backbone. GPT-OSS-20B and GPT-OSS-120B use a broad portion of the 1–10 scale, assigning nontrivial mass across the range and providing a usable dynamic range for ranking. In contrast, Qwen3-30B-A3B and especially Qwen3-235B-A22B exhibit a strong ceiling effect, with scores heavily concentrated near 10. This saturation suggests overconfident scoring and reduces score-based discrimination among candidates.

## A.2. Prompt Sensitivity of LLM-Judges

The LLM-judge prompt used in our experiments are adapted from prior evaluation prompts used in Zhang et al. (2025b) and Phan et al. (2025). To test whether our results are an artifact of this specific prompt, we re-run LLM-judge scoring with two alternative templates. Following Ma et al. (2025), we adopt a 0–7 proof-grading prompt used (aka ProofGrader). Additionally we bring a binary correctness prompt from Nie et al. (2025) (aka UQ). For GPT-OSS-20B and GPT-OSS-120B, we score each candidate with 64 independent judge calls and average, then compare the induced rankings across prompts using Spearman correlation. The rankings are highly consistent: $\rho = 0.961/0.954$ (ours vs. ProofGrader), $\rho = 0.938/0.950$ (original vs. UQ), and $\rho = 0.912/0.915$ (ProofGrader vs. UQ) for GPT-OSS-20B/120B, respectively. These correlations ($> 0.9$ throughout) indicate that while prompts change score scales, they have limited effect on relative ordering.

## A.3. Token Count: CBU VS. LLM-Judges

Table 5 compares inference cost and sampling diversity between LLM-judges and CBU. The average token usage per generation is comparable across methods, with CBU staying within ($\pm$ 15%) of the judge across backbones (e.g., +1.3% on Qwen3-235B, +15.0% on Qwen3-30B, +9.5% on GPT-OSS-120B, and (-7.4%) on GPT-OSS-20B). To quantify diversity across repeated rollouts, we embed each generation with Gemini Embedding 001 (Lee et al., 2025b) and compute the mean pairwise cosine similarity. CBU yields slightly lower similarity than LLM-judge across all backbones (typically by 0.005–0.008), indicating modestly higher variation across rollouts, although both methods remain highly similar overall (cosine ($\approx$ 0.96-0.97)).

*Table 5.* Token counts and pairwise cosine similarity statistics (mean $\pm$ std [min, max]) across generations.

| Models | LLM-Judge | | Utility | |
|---|---|---|---|---|
| | # Tokens | Cosine Similarity | # Tokens | Cosine Similarity |
| Qwen3-235B-A22B | 12152.53 | $0.969 \pm 0.008$ [0.910, 0.992] | 12310.33 | $0.964 \pm 0.010$ [0.896, 0.991] |
| Qwen3-30B-A3B | 10352.32 | $0.972 \pm 0.008$ [0.917, 0.992] | 11903.18 | $0.964 \pm 0.010$ [0.900, 0.991] |
| GPT-OSS-120B | 2905.85 | $0.972 \pm 0.007$ [0.915, 0.991] | 3182.76 | $0.964 \pm 0.013$ [0.896, 0.989] |
| GPT-OSS-20B | 5610.09 | $0.963 \pm 0.009$ [0.886, 0.988] | 5196.76 | $0.957 \pm 0.016$ [0.868, 0.987] |

# B. Evaluation Metrics

*Table 6.* **Formal definitions of evaluation metrics.** Here $\pi_k$ denotes the index of the $k$-th ranked candidate, $y^{(\pi_k)} \in \{0, 1\}$ is its correctness label, $\mathcal{C}$ and $\mathcal{W}$ are the sets of correct and wrong candidates, $s(\cdot)$ is the scorer, and $H$ is the human-written solution.

| Metric | Definition |
|---|---|
| Top1 accuracy (Acc@1) | $y^{(\pi_1)}.$ |
| Recall@5 | $\dfrac{1}{|\mathcal{C}|} \displaystyle\sum_{k=1}^{5} y^{(\pi_k)}.$ |
| AUC (pairwise separation) | $\dfrac{1}{|\mathcal{C}||\mathcal{W}|} \displaystyle\sum_{c \in \mathcal{C}} \sum_{w \in \mathcal{W}} \left( \mathbf{1}\{s(C_{\text{good}}) > s(C_{\text{wrong}})\} + \tfrac{1}{2}\mathbf{1}\{s(C_{\text{good}}) = s(C_{\text{wrong}})\} \right).$ |
| HumanWin | $\mathbf{1}\left\{ s(H) > \dfrac{1}{|\mathcal{W}|} \displaystyle\sum_{w \in \mathcal{W}} s(C_{\text{wrong}}) \right\} + \tfrac{1}{2}\mathbf{1}\left\{ s(H) = \dfrac{1}{|\mathcal{W}|} \displaystyle\sum_{w \in \mathcal{W}} s(C_{\text{wrong}}) \right\}.$ |
| MeanWin | $\mathbf{1}\left\{ \dfrac{1}{|\mathcal{C}|} \displaystyle\sum_{c \in \mathcal{C}} s(C_{\text{good}}) > \dfrac{1}{|\mathcal{W}|} \displaystyle\sum_{w \in \mathcal{W}} s(C_{\text{wrong}}) \right\} + \tfrac{1}{2}\mathbf{1}\left\{ \dfrac{1}{|\mathcal{C}|} \displaystyle\sum_{c \in \mathcal{C}} s(C_{\text{good}}) = \dfrac{1}{|\mathcal{W}|} \displaystyle\sum_{w \in \mathcal{W}} s(C_{\text{wrong}}) \right\}.$ |

# C. Reproducibility

All codes used throughout the paper, and parsed generation results are included in the supplementary results file for submission.

# D. Details on EXPERTMATH .

EXPERTMATH comprises 192 expert-written mathematics problems and 425 LLM-generated problems derived from RealMath. We plan to release the 425 LLM-generated problems on Hugging Face shortly; the 192 expert-written problems remain under embargo until after July 2026 due to requirements of the funding body. During the embargo, we will provide evaluation on EXPERTMATH for submitted models upon request. Below, we provide expert-written questions and solution pairs (Section 4), along with LLM-generated questions (Section 7).

---

**Expert-Written: Grassmannian Trace (Q&A)**

**Question.**
Consider the Grassmannian $Gr(n, 2n)$ of $n$-planes in $\mathbb{C}^{2n}$. Let $R = \bigoplus_{m=0}^{\infty} R_m$ denote the (homogeneous) coordinate ring of $Gr(n, 2n)$, so that $R_m$ is the degree-$m$ part. Let $B$ be a symplectic form on $\mathbb{C}^{2n}$. The map $\varphi_B : Gr(n, 2n) \to Gr(n, 2n)$ which sends each subspace $U$ to $U^{\perp}$, its orthogonal complement with respect to $B$, is an automorphism of $Gr(n, 2n)$ (as an algebraic variety). Hence, via pull-back, it induces an automorphism, also denoted $\varphi_B$, on the ring $R$ and on each homogeneous component $R_m$. What is the trace of $\varphi_B$ acting on $R_m$, when $n = 3$ and $m = 2$?

**Solution.**
This is the same as the number of symmetric (i.e., transpose-invariant) plane partitions of shape $n \times n$ and entries in $\{0, 1, \ldots, m\}$, as discussed in [K] and [H]. The number 35 can then be computed from the famous product formula

$$\prod_{i=1}^{n} \frac{2i + m - 1}{2i - 1} \prod_{1 \le i < j \le n} \frac{i + j + m - 1}{i + j - 1}$$

for symmetric plane partitions.

Another way to do it is directly with the standard monomial basis. Recall that, in general for the Grassmannian $Gr(a, a+b)$ of $a$-planes in $\mathbb{C}^{a+b}$, and its coordinate ring $R = \bigoplus_{m=0}^{\infty} R_m$, a basis of $R_m$ is indexed by the semistandard Young tableaux (SSYT) of shape $m^a$ (i.e., $m$ columns of length $a$), with entries in $\{1, \ldots, a+b\}$: we associate to each such tableau $T$ the product of the Plücker coordinates corresponding to the columns of $T$. In the case when $a = b = n$, the automorphism $\varphi_B$ acts as a permutation on this basis in the following way: we replace each column by its reflected complement in $\{1, \ldots, 2n\}$ (where reflected complement means we first replace each $i$ by $2n + 1 - i$, and then take the complement).

So, without using plane partitions, the problem boils down to counting $3 \times 2$ SSYT whose columns are invariant under the reflected complement operation, which can also be directly counted to be 35.

---

**Expert-Written: PGL$_2$ Deligne–Lusztig (Q&A)**

**Question.**
Let $k$ be a local non-Archimedean field with integers $O_k$, uniformizer $t$, and residue field $\mathbb{F}_q$ of characteristic $p$. Let $\breve{k}$ be the completion of the maximal unramified extension of $k$, and let $O_{\breve{k}}$ be its integers. Consider an unramified reductive group $G$ over $k$. Suppose $T$ is an unramified torus of $G$ and let $U$ be the unipotent radical of a Borel subgroup containing $T$ and defined over $\breve{k}$.

Fix a parahoric model $\mathcal{G}$ of $G$ over the integers $O_k$. Fix an integer $r \ge 1$. We may consider $G_r := \mathcal{G}(O_{\breve{k}}/t^r O_{\breve{k}})$ as a perfect algebraic group (of perfectly finite type) over the residue field $\mathbb{F}_q$ by using the positive loop functor (also called jet scheme) construction. The Frobenius $F$ of $\breve{k}/k$ acts naturally on $G_r$. Also, the closures of $T, U$ in $\mathcal{G}$ define by the same procedure subgroups $T_r, U_r$ of $G_r$ (with $T_r$ being $F$-stable but $U_r$ not). Let

$$X_r = \{g \in G_r : g^{-1}F(g) \in U_r\}.$$

It admits an action of $(G_r)^F \times (T_r)^F$ by $(g, t) : x \mapsto gxt$. For a character $\phi$ of $(T_r)^F$, let $R_{T,U,r}(\phi)$ denote the alternating sum of the $\phi$-isotypic parts of the $\ell$-adic étale cohomology groups of $X_r$, regarded as a virtual $(G_r)^F$-module (very similar to classical Deligne–Lusztig theory).

Now, assume that we are in the very special case $G = PGL_2$, $T$ a split torus, $U$ is $k$-rational, $\mathcal{G}$ hyperspecial, $r$ arbitrary, and $\phi = 1$ the trivial character. Compute the number of different irreducible representations appearing in $R_{T,U,r}(1)$.

---

**Solution.**

As $U$ is $k$-rational, $U_r$ is $F$-stable. It then follows that $X_r$ is just a disjoint union of translates of $U_r$, indexed by $(G_r/U_r)^F$. As $U$ is (the perfection of) an affine space, it only contributes to the cohomology by a degree shift. Thus, inserting $\phi = 1$, we see

$$R_{T,U,r}(1) = \mathrm{Ind}_{B_r^F}^{G_r^F}(1),$$

the induction of the trivial representation of $B_r^F$ to $G_r^F$ (it sits in one cohomological degree). The number of irreducible components of this representation is then given by the Mackey formula, which computes

$$\dim_{G_r^F}\big(R_{T,U,r}(1), R_{T,U,r}(1)\big) = \Big\langle \mathrm{Ind}_{B_r^F}^{G_r^F}(1), \, \mathrm{Ind}_{B_r^F}^{G_r^F}(1) \Big\rangle_{G_r^F}$$

(inner product on class functions of the finite group $G_r^F$) in terms of the double cosets $B_r^F \backslash G_r^F / B_r^F$.

An easy calculation with matrices in $G_r = PGL_2(O_k/t^r O_k)$ shows that there are precisely $r+1$ different double cosets, represented by

$$\begin{pmatrix} 1 & 0 \\ p^i & 1 \end{pmatrix} \quad (1 \le i \le r), \qquad \text{and} \qquad \begin{pmatrix} 0 & 1 \\ 1 & 0 \end{pmatrix}.$$

Hence

$$\Big\langle \mathrm{Ind}_{B_r^F}^{G_r^F}(1), \, \mathrm{Ind}_{B_r^F}^{G_r^F}(1) \Big\rangle_{G_r^F} = r + 1.$$

(Cf. [DI, Example 3.2.1] for the case $G = GL_2$ and $r = 2$.)

---

**Expert-Written: Auslander–Reiten Translate (Q&A)**

**Question.**

Let $G$ be the elementary abelian group of order 9 and $K$ the field with 3 elements. Let $KG$ be the group algebra of $G$ over $K$. Let $S$ be the unique (up to isomorphism) simple $KG$-module. What is the vector space dimension of $\tau^4(S)$, the Auslander–Reiten translate applied four times to $S$?

**Solution.**

$KG$ is isomorphic as a $K$-algebra to $K[x,y]/(x^3, y^3)$ since $K$ has characteristic 3. Now $KG$ is a symmetric Frobenius algebra and thus the Auslander–Reiten translate $\tau$ is isomorphic to $\Omega^2$, the second syzygy functor, in the stable module category. Thus one has to calculate the eighth syzygy module of $S$, which is reduced to standard commutative algebra/linear algebra.

---

**Expert-Written: Varchenko–Gelfand Ideal (Q&A)**

**Question.**

Let $D_4$ be the reflection arrangement. What is the smallest degree in which the (filtered) Varchenko–Gelfand ideal of this arrangement can be generated? (That is, what is the smallest degree $d$ such that the Varchenko–Gelfand ideal can be generated with relations of degree $d$ or less.)

**Solution.**

The twelve hyperplanes of the $D_4$ arrangement are given by the following equations

$x_1 - x_2, \; x_1 + x_2, \; x_1 - x_3, \; x_1 + x_3, \; x_1 - x_4, \; x_1 + x_4, \; x_2 - x_3, \; x_2 + x_3, \; x_2 - x_4, \; x_2 + x_4, \; x_3 - x_4, \text{ and } x_3 + x_4.$

This arrangement has 124 circuits, defining 124 generators of the Varchenko–Gelfand ideal. The 124 circuits are

$$\{\{0,1,2,3\}, \{0,1,4,5\}, \{2,3,4,5\}, \{0,6,2\}, \{1,6,3\}, \ldots, \{8,9,10,11\}\}$$

This already tells us that $2 \le d \le 4$, so we just need to check the smallest degree of generation. Let us start with an easy check for $d = 2$: there are 16 relations that have size three or less. We can construct a "truncated" Varchenko–Gelfand ideal using only the circuits of size three or less. If this generates the whole ideal, then we will be done.

This smaller ideal is generated by

$$e_0^2 - e_0, \; e_1^2 - e_1, \; \ldots, \; e_6 e_9 - e_6 e_{11} + e_9 e_{11} - e_9$$

We can check that this smaller ideal contains the bigger one. Since the smaller ideal is generated by degree-two elements, $d = 2$.

## RealMath: Recurrences over $\mathbb{F}_3$

**Question.**
Let $\mathbb{F}_q$ be a finite field with $q = 3$. Consider linear recurrence relations of order exactly $k = 6$. A sequence $s_0, s_1, \ldots, s_{11}$ is determined by a characteristic polynomial $P(x)$ of degree 6 (with non-zero constant term) and initial values. How many such sequences of length 12 exist such that the underlying minimal polynomial has degree exactly 6?

## RealMath: Simplex Shape Ratio

**Question.**
Let $n = 6$. The dimensionless shape ratio for a regular simplex is

$$\mathcal{I}_n = \frac{S^n}{V^{n-1}},$$

where $S$ is the total surface area and $V$ is the volume.
For a regular simplex, one (incorrect) simplification is sometimes written as

$$\mathcal{I}_n = \frac{n^n (n+1)^{(n+1)/2}}{\sqrt{n!}} \quad \text{(distractor; derive the correct one).}$$

Instead, use the known relation for a regular simplex:

$$\frac{S^n}{V^{n-1}} = n^n (n+1)^{(n-1)/2} (n!).$$

Calculate this value for $n = 6$.

## RealMath: Lattice Path Constraint

**Question.**
Let $L$ be the set of lattice paths from $(0,0)$ to $(15, 12)$ taking steps $E = (1, 0)$ and $N = (0, 1)$ such that the path never touches or rises above the line

$$y = \frac{4}{5} x$$

after the origin. Calculate the size of $L$.

# E. Prompts.

In this section, we list the prompts used throughout the paper:

1. Consequence-Based Utility Prompt (Section 5)
2. LLM-Judge: Default Prompt (Section 5)
   (a) LLM-Judge: ProofGrader Prompt
   (b) LLM-Judge: UQBench Correctness Prompt
3. Problem Generation: RealMath (2) (Section 7)
4. Error Analysis Prompt (Section 6)

---

**(1) Consequence-Based Utility Prompt**

```
{original question}
```

```
{candidate solution}
```
Refer to the question-solution set provided above. Solve the provided question below and output the final answer in the following format: $\boxed{N}$ .

```
{variant question}
```

---

**(2) LLM-Judge: Default Prompt**

You are an impartial mathematical judge.
You will be given a math problem and a proposed solution.
The solution may or may not be correct and does not explicitly state a final answer.

Your task is to carefully evaluate the solution for logical correctness, mathematical validity, completeness, and rigor, with special emphasis on whether the reasoning fully and correctly solves the given problem.

You must independently reason through the problem first, forming your own reference solution or partial verification, and then compare the given solution against that reasoning.

**Evaluation Instructions:**

1. Assess the solution step by step.

2. Verify all mathematical claims, derivations, and logical transitions.

3. Identify any gaps, unjustified steps, incorrect assumptions, or missing arguments.

4. Consider whether the solution fully resolves the question as stated.

5. Partial or high level arguments are insufficient unless explicitly justified.

**Scoring Rubric:** Assign a single integer score from 1 to 10.

• **10**: Completely correct, rigorous, logically sound, and fully solves the problem.

• **9**: Essentially correct with very minor omissions that do not affect correctness.

• **7–8**: Mostly correct but with minor logical gaps or unclear justifications.

• **5–6**: Partially correct but missing key arguments or containing nontrivial ambiguity.

• **3–4**: Significant errors or omissions, but some relevant ideas are present.

• **1–2**: Largely incorrect with major logical or mathematical flaws.

• **1**: Completely incorrect or irrelevant.

**Required Output Format:**

```
Summary:
<brief neutral summary of your evaluation>

Detailed Analysis:
<concise but precise discussion of correctness, gaps, or errors>

Score: <integer from 1 to 10>
```

**Question**
```
{original question}
```
**Solution**
```
{candidate solution}
```

## (2-a) LLM-Judge: ProofGrader Prompt

You are an **expert math proof grader**. You are judging the correctness of an LLM-generated proof for a math problem.

**Input**
Your input will consist of:

- **Problem Statement**: A mathematical problem that the proof is attempting to solve.

- **Proof Solution**: The proof that you need to evaluate. This proof may contain errors, omissions, or unclear steps. The proof was generated by another language model.

**Task**
Analyze the proof carefully.

- Identify logical errors, incorrect steps, or unclear reasoning.

- Give an **integer** score between 0 and 7 with a brief overall assessment.

**Output Format**
Respond with **only** well-formed XML using the structure below.
Do not include any extra text or Markdown.

**Requirements:**

- `<score>` must be an integer in `[0, 7]`.

- `<assessment>` must be a **detailed analysis** that explains your reasoning step-by-step and provides a clear **rationale for the score**. Reference specific claims/lines if present.

- `<errors>` must be a list of specific issues (empty if score = 7).

**Example output:**

```
<score>0</score>
<assessment>The proof shows a good understanding of the main idea, but has
some unclear reasoning and minor mistakes...</assessment>
<errors>
1. specific error 1,
2. specific error 2,
...
</errors>
```

**Scoring Guidelines (0–7 scale)**

- **0**: Completely incorrect; proof is irrelevant, nonsensical, or shows no understanding.

- **1–2**: Very poor; major logical flaws, does not solve the problem, but may contain fragments of relevant reasoning.

- **3–4**: Partial progress; captures some correct reasoning or key ideas, but has significant logical errors, missing steps, or incomplete arguments that make the proof invalid overall.

- **5–6**: Largely correct; the proof is overall valid and reaches the correct conclusion. Contains only **minor issues** (e.g., small calculation mistakes, notation slips, or slightly unclear wording) that do not undermine correctness.

- **7**: Fully correct; the proof is complete, logically sound, and clearly presented with no substantive errors.

---

**Problem Statement**
{original question}
**Proof Solution**
{candidate solution}

---

### (2-b) LLM-Judge: UQBench Correctness Prompt

Please act as an impartial judge and evaluate whether the AI assistant's response is completely correct in both process and conclusion. Consider correctness, usefulness, completeness, and depth in your assessment. Consider whether this answer completely solves the user's question. You should rely on your own reasoning to form a reference or partial solution first and compare the AI assistant's response to your reasoning. Begin your evaluation by giving a brief summary of your thoughts on the response. Focus on whether it is accurate, addresses the question well, and is reasonably detailed. Be precise about any errors or gaps you notice. Keep your explanation unbiased and do not let any external factors or the question's difficulty level sway your evaluation.

**Notes:**

1. If the answer is partial, high-level, or just states that this is an open problem, you should not accept it.

2. If the answer lacks details or is not comprehensive, you should not accept it.

3. If the answer contains any errors, you should not accept it.

4. You should only accept the answer if it is at least 95%.

5. If the question is a puzzle, the requirement of completeness can be appropriately relaxed.

After providing your explanation, please decide whether this answer is the correct answer to the question. Think twice about whether this answer solves the user's question.

You must strictly follow this format: `Accepted: [[Y]]` if you decide to accept the answer or `Accepted: [[N]]` if you decide not to accept the answer.

---

**[Question]**
**Question Content**
{original question}

**[The Answer to Evaluate]**
{candidate solution}

---

### (3) Problem Generation: RealMath(2)

You will be given a math problem. I want to train a student by practicing the sub-skills needed to solve it.

Create 5 standalone practice problems (each should be meaningful on its own) such that mastering them makes the original problem straightforward.

**Constraints:**
- Do **NOT** produce trivial "split the original into parts" questions or simple plug-in/replace-numbers variants.

- Each problem must target a specific sub-skill needed for the original.

- Across the 5 problems, cover all key sub-skills without heavy overlap.

- Keep the difficulty as hard as possible.

- Each question should have an **INTEGER** answer bigger than 1000.

Return in the following format.

```
<description>
{description of the five problems}
</description>

<questions>
["...","..."] {a Python list of the questions}
</questions>
```

## (4) Error Analysis Prompt

You are a mathematical reasoning auditor.

You will be given:

1. a mathematical question, and

2. an LLM-generated solution.

Your goal is to tag ONLY the clearest, high-confidence reasoning failures in the solution.
Be conservative: if a category is only weakly suggested or depends on subtle theory, do NOT tag it.

You may output ZERO, ONE, or MULTIPLE categories from the list below.

**Failure categories (tag only when strongly supported by the text):**

1. Incorrect reasoning

2. Unjustified Compression

3. Unjustified Interpretation

4. External References

**Quoting rules (STRICT):**
- For every tagged category, provide 1–3 verbatim quotes from the solution.

- Each quote must be an exact substring of the solution text.

- For EACH quote, provide a detailed explanation:

  1. what the quote claims,

  2. why it is wrong / unjustified / drifting,

  3. what would be needed to fix it.

Do NOT use external sources to validate math facts.
Do NOT judge the final answer directly.

**Output format (STRICT JSON):**

```
{
  "categories": [
    {
      "id": <1-6>,
      "name": "<category name>",
      "evidence": [
        {
          "quote": "<verbatim quote 1>",
```

```
          "analysis": {
            "claim": "...",
            "why_problematic": "...",
            "what_needed": "..."
          }
        },
        {
          "quote": "<verbatim quote 2>",
          "analysis": {
            "claim": "...",
            "why_problematic": "...",
            "what_needed": "..."
          }
        }
      ]
    }
  ]
}
```

If no category applies:

```
{ "categories": [] }
```

**Input format you will receive:**

```
[QUESTION]
...question text...

[SOLUTION]
...solution text...
```

**Sample output illustrating multiple quotes inside one category:**

```
{
  "categories": [
    {
      "id": 4,
      "name": "Structural opacity / representation drift",
      "evidence": [
        {
          "quote": "The roots of type D4 are given by the set of vectors:
          {± e_i ± e_j ... }",
          "analysis": {
            "claim": "Defines the ground set as a signed root set with multiple
            vectors per hyperplane normal direction.",
            "why_problematic": "Later steps treat these vectors as indexing
            hyperplanes without stating the identification ±a defining the same
            hyperplane. This changes what the variables represent and what
            counts as a circuit.",
            "what_needed": "State explicitly whether the ground set is
            hyperplanes, positive roots, or roots modulo ±, and map roots
            to hyperplanes before forming the polynomial ring variables."
          }
        },
        {
          "quote": "The number of such roots (and thus hyperplanes) is
```

```
            N = ... = 12.",
            "analysis": {
              "claim": "Equates the number of roots to the number of hyperplanes
              and gives N=12.",
              "why_problematic": "Given the earlier signed root set, the root
              count and hyperplane count differ unless an identification
              is declared. The argument relies on N to define the
              polynomial ring variables, so this mismatch affects the rest
              of the reasoning.",
              "what_needed": "Clarify the counting convention and rewrite N
              consistently (either count roots as 24 or explain why hyperplanes
              are 12)."
            }
          }
        ]
      }
    ]
  }
}
```

