# OpenReview forum: "Judging What We Cannot Solve: A Consequence-Based Approach for Oracle-Free Evaluation of Research-Level Math"
_ICML.cc/2026/Conference — ICML 2026 spotlight_

### Official Review · Reviewer_qNMi · 2026-03-12

**Soundness:** 2
**Presentation:** 4
**Significance:** 3
**Originality:** 3
**Overall Recommendation:** 4
**Confidence:** 3

**Summary:**

This paper proposes a novel oracle-free evaluation framework named Result-Based Utility (CBU), designed to assess the generation quality of large language models on research-level mathematical problems where standard answers are unavailable. The method is grounded in the assumption that high-quality candidate solutions should contain sufficient methodological information, such that when used as in-context examples, they improve the model’s solving accuracy on related Neighborhood Questions. The authors construct the EXPERTMATH dataset and demonstrate that CBU outperforms existing generative reward models (GenRM) and LLM-Judges in ranking quality. Furthermore, the paper shows that CBU can more reliably penalize fake solutions containing logical flaws and unwarranted compression.

**Compliance With Llm Reviewing Policy:**

Affirmed.

**Final Justification:**

I recommend accepting this paper. After comprehensively evaluating the quality of the original manuscript and the authors’ rebuttal, I believe this paper meets the publication standards and demonstrates sufficient academic contributions with clear expression.

**Key Questions For Authors:**

W1：The paper aims to address the evaluation bottleneck of LLMs on “unknown research problems”. When facing a completely new mathematical conjecture with no supporting literature, how can the system automatically construct mathematically valid and verifiable “neighborhood questions Q” without relying on human expert prior knowledge?
W2：The paper states that the effectiveness of CBU heavily depends on the difficulty of neighborhood questions: if Q is too easy, the model can solve it without the candidate solution; if too hard, it cannot solve it even with hints. In a practical automated evaluation pipeline, how can the system adaptively ensure that generated Q falls within this valid difficulty range? If the difficulty distribution of Q cannot be stably controlled, will CBU scores suffer from excessive variance and lose reliability?
W3：The candidate solution is directly used as an in-context example, and the outcome-based reward is determined by whether downstream questions are solved correctly. However, larger LLMs tend to rely on “shortcuts” rather than genuine generalization during in-context learning (ICL). If an incorrect solution contains numerous correct background axioms, the LLM can easily exploit these knowledge shortcuts to produce apparent understanding and solve neighborhood questions. How do you prove that CBU evaluates the reasoning correctness of the solution, rather than merely the usefulness of background knowledge?

**Limitations:**

Although this paper presents an elegant thought experiment, its core evaluation mechanism contains irreconcilable logical flaws.
1. The method suffers from a severe “oracle transfer paradox” in fully unknown scientific discovery settings and cannot truly escape reliance on known answers.
2.The experimental design is significantly unfair: the authors use 64× more computational power for multi-round downstream sampling to outperform baseline reward models based on single-round evaluation.
3. Without disentangling and defending against in-context shortcut learning and false positives, the statistical and causal validity of the proposed “utility” metric is severely undermined.

**Strengths And Weaknesses:**

Strengths:
1.The main strength of the paper lies in its insightful reflection and qualitative analysis of current evaluation paradigms for large models.
2.The authors astutely identify the pain point of “validation difficulty” in complex mathematical reasoning, and provide detailed error analysis showing that existing LLM-Judges are easily deceived by authoritative citations and superficial logical coherence.
3.Additionally, translating the epistemological concept of support by consequences in mathematics into a concrete evaluation algorithm represents a highly inspiring interdisciplinary thought experiment.
Weaknesses:
1.However, the method suffers from two critical flaws: the oracle transfer paradox, whereby the system cannot automatically construct verifiable “neighborhood questions” for truly unsolved open scientific problems without relying on expert prior knowledge, rendering the approach inapplicable in real research scenarios.
2.The second flaw is the risk of false positives caused by shortcut learning, as CBU directly feeds candidate solutions as in-context exemplars to the model—making LLMs highly susceptible to being biased by correct background theorems piled into incorrect solutions, which can lead to artificially high scores.
3.The paper completely fails to causally disentangle the confounding effects of this information leakage.

---

> ### Author Rebuttal · Authors · 2026-03-31
>
> Dear Reviewer qNMi,
>
> Thank you for your time and effort in reviewing our paper.
> We have summarized our response to the 2 weaknesses,  3 questions and 2 limitations,  as W1-W2, Q1-Q3, and L1-L2.
>
> ---
>
> ## [W1 & Q1 & L1] Oracle Transfer Paradox
>
> We thank the reviewer for raising this edge case. In the paper, we consider three sources of neighborhood questions: expert-written questions, LLM-generated variants, and questions derived from ancestor literature. We agree that a conjecture with no supporting literature whatsoever is possible in principle, and in such a case, CBU may be less effective. However, we believe this regime is extremely rare in mathematics. Even the standard examples of major conceptual change still had identifiable precursors: Newton-Leibniz calculus grew out of the precalculus work of Cavalieri, Fermat, and Barrow, and Riemannian geometry grew out of Gauss’s intrinsic geometry of surfaces. Thus, even when the exact conjecture is new, there are usually prior concepts, techniques, or nearby questions from which meaningful neighborhoods can be constructed. We therefore expect ancestor-based neighborhood generation to apply in most realistic cases. More importantly, the fully literature-free case is difficult not only for CBU but for essentially any rapid evaluation method, including human judgment, since reliable assessment would itself require substantial time to absorb and validate the new framework.
>
> If, however, the reviewer has specific cases in mind, and thinks that papers with no earlier literature is more common than what we (authors) think, please feel free to share it with us. We would be happy to review the paper and think of other scalable ways to supplement our method.
>
> ---
>
> ##  [W2 & Q3 & L2] Robustness to false positives or “partially correct solutions”
> To start with, Section 6 and Figure 4 partially covers this issue. In Section 6 we work with four PhD students to conduct a qualitative study on areas CBU outperform judges. Among 112 trajectories studied 77 of them (approx. 68.8%) included partially wrong reasoning. LLM-Judges have failed to identify such errors, while CBU correctly penalizes them. This shows that **CBU is more effective in identifying and penalizing partially-correct solutions compared to LLM-Judges.**
> Additionally, we are running additional experiments with LLM-generated partially wrong solutions and will be adding them shortly during the review period.
>
> ---
>
> ## [Q2] Sensitivity to problem difficulty
> As the reviewer mentioned, if Q is too easy, the model can solve it without the candidate solution and CBU fails. However, as questions get harder, CBU actually improves in performance. We report this result in Figure 3 and lines 295 - 324. To summarize, in the easier regime LLM-Judges are a better option than CBU, while in the harder regime CBU performs better than LLM-Judges.
>
> Accordingly, for practitioners who plan to use CBU as a part of an automated pipeline we recommend to aim for possibly the most difficult neighborhood questions possible (as long as they are easier than the original question they try to evaluate).
>
> ---
>
> Please feel free to ask any additional questions, and consider raising the score if our explanations helped you resolve some concerns about our paper.

---

> > ### Author Rebuttal · Reviewer_qNMi · 2026-04-04
> >
> > Thank you for your response. The issues have been resolved, and I have decided to maintain my original score.

---

> > > ### Author Response · Authors · 2026-04-04
> > >
> > > Hi, Thanks for your acknowledgement. Here are our additional results on Partially Correct Responses. Feel free to have a look if this can help resolve any concerns further!
> > >
> > > ## Additional Results on Partially Correct Responses
> > >
> > > Here are our additional results on partially-correct responses. However, constructing such examples is nontrivial for graduate- or research-level tasks. We therefore build controlled hard negatives from verified correct human or strong-model solutions in the gold set: we truncate each solution at its 50% point, use Qwen3-4B-Thinking to complete the remaining reasoning, and retain only continuations whose final answer is incorrect. This produces examples that preserve substantial correct intermediate structure while remaining wrong overall. We then score these continuations with CBU and with LLM judges. Since CBU is natively in ([0,1]) while judge scores are on a 0 to 10 scale, we normalize judge scores by dividing by 10, so that 0 is the ideal value for both metrics on this benchmark. Under this construction, every evaluated continuation is incorrect, so the desired behavior is to assign scores as close to 0 as possible.
> > >
> > > **On this stress test, CBU is substantially more conservative than LLM judges. Across both GPT-OSS-20B and GPT-OSS-120B, CBU has median 0.0, whereas normalized LLM judges have medians 0.240 and 0.205; moreover, CBU assigns the lower score on 91.3% and 86.2% of paired examples, respectively.** All paired significance tests strongly reject equality. These results suggest that, CBU is much less prone than LLM judges to over-credit partially correct but ultimately incorrect solutions.
> > >
> > > All numbers below use normalized LLM judge scores, obtained by dividing raw 0 to 10 judge outputs by 10. Lower is better because every continuation is incorrect.
> > >
> > > ### Results on incorrect continuations
> > >
> > > | Backbone     | (n) |  CBU mean | LLM judge mean | CBU median | LLM judge median |
> > > | ------------ | --: | --------: | -------------: | ---------: | ---------------:|
> > > | GPT-OSS-20B  | 849 | **0.070** |          0.301 |  **0.000** |            0.240 |
> > > | GPT-OSS-120B | 835 | **0.097** |          0.276 |  **0.000** |            0.205 |
> > >
> > > ### Paired comparison
> > >
> > > | Backbone     | Mean paired diff (judge - CBU) | Median paired diff | CBU lower on | Wilcoxon (p) | Paired (t)-test (p) |
> > > | ------------ | -----------------------------: | -----------------: | -----------: | -----------: | ------------------: |
> > > | GPT-OSS-20B  |                          0.231 |              0.214 |        91.3% |   9.276e-102 |          2.591e-130 |
> > > | GPT-OSS-120B |                          0.179 |              0.195 |        86.2% |    1.131e-70 |           2.477e-68 |
> > >
> > > ---
> > >
> > > **Taken together, these results show that even on wrong responses containing substantial correct intermediate structure, CBU suppresses scores much more reliably than LLM judges.**
> > >
> > > Thank you again for your time to look at our paper. Please feel free to remind us if we have missed something. Also, if we have handled all your concerns, please consider raising the score.

---

### Official Review · Reviewer_TAAQ · 2026-03-12

**Soundness:** 2
**Presentation:** 2
**Significance:** 3
**Originality:** 3
**Overall Recommendation:** 4
**Confidence:** 3

**Summary:**

This work is about judging math problems that are too hard for machines to fully solve. Standard LLM judges often get tricked by plausible-sounding but wrong solutions, and experts are expensive. The authors introduce Consequence-Based Utility, which scores a candidate by seeing how much it helps solve similar, verifiable math questions. Essentially, it’s testing the “side effects” of a solution rather than the solution itself. They test this on EXPERTMATH, which contains very difficult research-level problems. CBU consistently outperforms LLM judges and reward models, especially in the hardest problems where models themselves fail. The authors proceed to analyze the concept of transfer via neighborhood questions, and the paper provides visualizations showing how CBU better separates correct from wrong answers. The authors strive to present a central context for why evaluating downstream utility is more reliable than just inspecting reasoning traces, and it seems to work pretty well for subtle errors and edge cases that trip up other methods

**Compliance With Llm Reviewing Policy:**

Affirmed.

**Final Justification:**

I suggest that this paper be accepted. After carefully assessing the original manuscript's quality as well as the authors' reply, I think this work satisfies publishing requirements and shows adequate contributions that are expressed clearly.

**Key Questions For Authors:**

Would it be possible to quantify the impact of using automatically generated versus expert-written neighborhood questions? Also, what happens if the solver itself has systematic biases that propagate into the neighborhood evaluation?

**Limitations:**

yes

**Strengths And Weaknesses:**

**Strengths**

- innovative evaluation that doesn’t rely on having a ground-truth solution

- works better than previous oracle-free methods on tough problems

- clear empirical evidence and well-explained methodology

- includes detailed practical guidance for implementing CBU

**Weaknesses**

- neighborhood question creation could be a bottleneck

- method relies on the solver being good enough to transfer learning to neighborhoods

- mostly focused on math; unclear how it generalizes

---

> ### Author Rebuttal · Authors · 2026-03-31
>
> Dear Reviewer TAAQ,
> Thank you for your time and effort in reviewing our paper. We have summarized our response to the 3 weaknesses and 1 question as W1-W3 and Q1.
>
> ---
>
> ## [W1] Trickiness to construct neighborhood questions
>
> In the paper, we consider three sources of neighborhood questions: expert-written questions (Section 4), LLM-generated variants, and questions derived from ancestor literature (Section 7.2). While expert-written questions are difficult to approach, the other two methods are reasonable. First, regarding the automated creation of questions by LLMs, several relevant work has already shown such. [1,2] We also retry ourselves in Table 4 and confirm that this is applicable in our setting. Second, we would like to highlight that it is very unlikely for one to encounter a scenario where either method is not possible. Conjectures with absolutely no previous work, are extremely rare in mathematics meaning our second method shell be usable in most cases.  Finally, **the most recommended area for CBU to be used is incremental-level math research**, the type of research which pushes an already existing domain further. **Our paper proves CBU to outperform traditional methods in such areas throughout Sections 4 to 5** by constructing our own dataset of faculty-authored research-level math and comparing it against LLM-Judges and GenRMs.
>
> [1] https://arxiv.org/abs/2505.12575
> [2] https://arxiv.org/abs/2602.24173v1
>
> ---
>
> ## [W2] Heavy reliance on good solvers
>
> As mentioned by the reviewer CBU does require a solver with enough capability on the neighborhood questions for downstream utility to be informative. However, this dependence is not unique to CBU. **LLM judges are also strongly backbone-dependent**, with Table 2 showing that their performance also scales in size. More importantly, our results suggest that **CBU mitigates this dependence rather than amplifying it**: for example, Qwen3-30B-A3B with CBU reaches 85.71 HumanWin and 76.24 AUC, substantially outperforming a much larger Qwen3-235B-A22B used as an LLM judge, which attains 67.14 HumanWin and 69.48 AUC. In addition, the paper shows that CBU preserves stronger correct-wrong separation than LLM judges even in the hardest regime, including cases where the underlying model often fails to solve the target question itself. Thus, CBU does not require a solver that can already solve the research problem; it requires only enough local competence on verifiable neighborhood problems for conditioning on a candidate solution to produce a measurable downstream effect. Since the method is explicitly designed for research-level questions that remain open to current LLMs, this is the intended operating regime rather than a hidden assumption. We will revise the paper to make this scope condition more explicit.
>
> ---
>
> ## [W3] Domain-specific limitation
>
> To start with, the authors mostly have focused on Math as the scope of this paper, additionally due to **limitations in budget we were unable to look for additional faculty members to help us in creating a multi-domain version of ExpertMath**. Another main reason why we focus on math is because of its dry-nature. **While most of the other fields in STEM (e.g., chemistry, biology, and physics) require observation and experiments to conduct research-level activity, math is free from such, and is most easily applicable** to our proposed method. We are also running experiments similar to that of Section 7 with datasets from other domains, and will share during the review period.
>
> ---
>
> ## [Q1] Comparison of expert-written and LLM-generated neighborhood questions
>
> Qualitatively, the authors experienced that this highly relies on how specific the “prompt” provided to expert / LLMs are. We’ve encountered cases where experts will submit almost-identical duplicates and we had to guide them to resubmit (as mentioned in lines 190-204). Additionally, when using questions derived from ancestor literature, we notice that some of these questions are totally unrelated with the original one. Probably because the LLM concentrated on a niche portion of the paper that does not overlap between the two papers. However, this was steerable via prompts. Other than such we do not notice a big qualitative difference in the quality of neighborhood questions. Rather, we notice that human-experts greatly outperform LLMs in the ability to create original questions. LLMs seem to greatly lack the ability to create meaningful research-level questions with everything properly defined. In most of our attempts, LLMs tried to use tricks , either by adding meaningless computation steps or using weird and unfamiliar notation , to trick the readers and arbitrarily try to make the question difficult.
>
> ---
>
> Please feel free to ask any additional questions, and consider raising the score if our explanations helped you resolve some concerns about our paper.

---

> > ### Author Rebuttal · Reviewer_TAAQ · 2026-04-02
> >
> > I have decided to maintain my original score, as I believe it already reflects a fair and sufficiently positive assessment of the work.

---

> > > ### Author Response · Authors · 2026-04-06
> > >
> > > Here are our further experiments on domain generalization of CBU. To probe transfer beyond math, during the rebuttal period we ran a preliminary version of Section 7.2 on FrontierScience and HLE-Verified. For each instance, we used Gemini-3-Pro to generate a closely related variant, added prompts to avoid leaking the original answer or the key domain fact, and kept only cases where Gemini-3-Pro, GPT-5-Pro, and Grok-4 agreed on the provisional answer. Using GPT-OSS-20B as the evaluator, we obtained the following results:
> > >
> > > | Benchmark       | Domain           | CBU (Ours) | LLM-Judge |
> > > | --------------- | ---------------- | ---------: | --------: |
> > > | FrontierScience | Chemistry        |       43.3 |      62.7 |
> > > | FrontierScience | Physics          |       72.8 |      70.3 |
> > > | HLE-Verified    | Biology/Medicine |       51.1 |      57.1 |
> > > | HLE-Verified    | Chemistry        |       53.3 |      62.8 |
> > > | HLE-Verified    | Physics          |       80.3 |      71.2 |
> > >
> > > These preliminary results suggest that CBU transfers better to reasoning-dominant domains than to knowledge-dominant ones. In particular, CBU outperforms LLM-Judge on physics in both benchmarks, but underperforms on chemistry and biology/medicine, where many items appear to depend more on access to specific knowledge than on transferable reasoning structure. **We therefore view CBU as primarily targeted at oracle-free, reasoning-heavy settings, especially mathematics and likely parts of physics, rather than as a universal evaluator for all STEM tasks.** It should be noted that this does not weaken the core claim of the paper, since the paper is about evaluation in research-level reasoning settings where standard answer-key-based evaluation is unavailable or insufficient. We will revise the paper to make this scope more explicit and to report these cross-domain results as preliminary evidence.
> > >
> > > ---
> > >
> > > We hope this response addresses your concerns. If so, we would appreciate your reconsideration of the score. If any questions or concerns remain, please let us know, and we will revise the response accordingly.

---

### Official Review · Reviewer_KgXY · 2026-03-12

**Soundness:** 3
**Presentation:** 3
**Significance:** 3
**Originality:** 3
**Overall Recommendation:** 5
**Confidence:** 3

**Summary:**

The paper proposes Consequence-Based Utility (CBU), a novel oracle-free evaluation method for research-level mathematical solutions. Traditional evaluation of complex math solutions often requires scarce expert validation or relies on LLM judges, which can be biased or unreliable. CBU addresses this by measuring how well a candidate solution improves a solver’s performance on a neighborhood of related, verifiable problems. Correct or near-correct solutions consistently transfer useful method-level information, enabling more reliable evaluation without access to ground-truth answers.

The authors construct EXPERTMATH, a dataset of 192 expert-written research-level math problems and 630 LLM-generated candidate solutions, and demonstrate that CBU outperforms LLM judges, reward models, and generative reward models across multiple backbones such as GPT-OSS-120B and Qwen3-235B. Qualitative analysis shows that CBU better penalizes incorrect reasoning, unjustified compression, and superficial stylistic cues. The paper also provides practical guidance for applying CBU, including neighborhood question construction and the number of rollouts needed for stable evaluation. Overall, this work introduces a principled and empirically validated approach for evaluating difficult, research-level problems without relying on ground-truth solutions.

**Compliance With Llm Reviewing Policy:**

Affirmed.

**Final Justification:**

The paper proposes a well-motivated and technically sound approach with strong empirical support, and the rebuttal further addresses concerns on generalization and robustness; I therefore support an Accept (5).

**Key Questions For Authors:**

1. Sensitivity to neighborhood construction

How sensitive is CBU to the choice of neighborhood problems? Are there guidelines or experiments showing how performance varies with different neighborhood questions?

2. Handling partially correct solutions

How does CBU treat solutions that are partially correct or contain useful intermediate steps? Are there scenarios where it might incorrectly penalize such solutions?

3. Computational cost and scalability

What is the computational overhead of running CBU compared to standard LLM-based judges, especially for large candidate pools or complex problem neighborhoods?

**Limitations:**

yes

**Strengths And Weaknesses:**

Strengths:
1. Clear motivation and technically sound
The paper presents a clear and well-motivated problem: evaluating research-level mathematical solutions without access to oracles or expert validation. The proposed Consequence-Based Utility (CBU) method is intuitive and technically sound, leveraging downstream performance on related, verifiable problems. Experiments are well-designed, including multiple LLM backbones and baselines, and the results convincingly support the claims.
2. Clear presentation and intuitive results
The paper is well-written, with a clear structure and coherent narrative. Figures and tables are informative, helping readers understand the evaluation process and outcomes.
3. Significant application in research-level problem solving
Addressing oracle-free evaluation for research-level mathematics is valuable, as it reduces reliance on scarce expert validation and biased LLM-judges. The approach could inform future work in evaluating complex problem-solving beyond standard benchmarks.

Weaknesses:
1. Domain-specific limitation
The method is primarily validated on research-level math problems, leaving its applicability to other domains untested. Broader generalization is unclear.
2. Dependence on neighborhood construction
The approach relies on the construction of a neighborhood of related problems, which could introduce design choices and affect stability. Guidelines are provided, but sensitivity analysis is limited.

---

> ### Author Rebuttal · Authors · 2026-03-31
>
> Dear Reviewer KgXY,
> Thank you for your time and effort in reviewing our paper. We have summarized our response to the 2 weaknesses and 3 questions as W1-W2 and Q1-Q3.
>
> ---
>
> ## [W1] Domain-Specific Limitation
>
> To start with, the authors mostly have focused on Math as the scope of this paper, additionally due to **limitations in budget we were unable to look for additional faculty members to help us in creating a multi-domain version of ExpertMath**. Another main reason why we focus on math is because of its dry-nature. **While most of the other fields in STEM (e.g., chemistry, biology, and physics) require observation and experiments to conduct research-level activity, math is free from such, and is most easily applicable** to our proposed method. We are also running experiments similar to that of Section 7 with datasets from other domains, and will share during the review period.
>
> ---
>
> ## [W2 & Q1] Sensitivity of Neighborhood Questions
>
> Table 2 and Table 4 may be used to explain the sensitivity for neighborhood questions. First, in Table 2 we can notice that generally MeanWin, the likelihood of a model to give higher scores the correct solutions (both humans and LLM written), is higher than HumanWin, the likelihood of a model to give higher scores to human-written solutions alone. Responses written by human experts are often terse and intuition-driven and are hard to understand compared to that of user-friendly LLM responses, which is likely to cause such a score gap. However, we would like to highlight that LLM-Judges suffer from the same issue. Additionally, as shown in the Table below, CBU is more effective that LLM-judges in evaluating the noisy human-written solutions.
>
> | Model              | LLM-Judge HumanWin | CBU HumanWin |
> |--------------------|-------------------|--------------|
> | Qwen3-235B-A22B    | 67.14             | 81.43        |
> | Qwen3-30B-A3B      | 47.14             | 85.71        |
> | GPT-OSS-120B       | 48.57             | 82.86        |
> | GPT-OSS-20B        | 52.86             | 74.29        |
>
> Additionally, in Table 4 despite judging for the same Real-Math dataset depending on how the neighborhood question was created the performance of CBU is affected to some extent. However in either method it outperforms LLM-Judges.
>
> To summarize, in this work we identify difficulty of neighborhood questions as a key driver in the utility of CBU, however in most of the cases observed (unless the question is too easy, which is out of the scope of this paper), CBU outperforms LLM-Judges in its ability to evaluate responses.
>
> ---
>
> ## [Q2] Partially Correct Solutions
>
> To start with, Section 6 and Figure 4 partially covers this issue. In Section 6 we work with four PhD students to conduct a qualitative study on areas CBU outperform judges. Among 112 trajectories studied 77 of them (approx. 68.8%) included partially wrong reasoning. LLM-Judges have failed to identify such errors, while CBU correctly penalizes them. This shows that CBU is more effective in identifying and penalizing partially-correct solutions compared to LLM-Judges.
>
> ---
>
> ## [Q3] Computational Overhead
>
> First, we would like to clarify that the main comparison in Table 2 is budget-matched across the scalable evaluators. Specifically, GenRM and LLM-Judge are each repeated 64 times to match CBU’s rollout budget, and all methods are given the same 16k-token reasoning allowance. Plain reward models are the only exception: we report them in their standard deterministic single-pass setting, which uses roughly 1/64 of the compute of the rollout-based methods. This is because plain reward models do not natively support test-time scaling mechanism as GenRM, LLM-Judge, and CBU. We view this as a limitation of plain reward models rather than an unfair advantage for CBU, and Table 2 reflects this clearly: plain RMs are far weaker in HumanWin, with Qwen2.5-Math-RM-72B at 1.63 and AceMath-72B-RM at 0.00, compared with 27.05 for Qwen3-235B-GenRM, 67.14 for Qwen3-235B-A22B as an LLM-Judge, and 81.43 for CBU. Importantly, the practical overhead of CBU is milder than the 64-rollout reference may suggest. Table 5 explicitly notes that CBU and LLM-Judge consume comparable numbers of tokens on average, so neither enjoys a systematic budget advantage; Figure 6 further shows that CBU converges at a similar rate overall, faster on some backbones, and that (n \ge 8) already keeps the mean normalized error below 0.05 across all tested models. Thus, while CBU is more expensive than a literal single-pass judge, its comparison to GenRM and LLM-Judge is fair in inference budget, and in practice a small-rollout regime already captures most of the benefit. We will make such points clearer in our paper to help the understanding of future readers.
>
> ---
>
> Please feel free to ask any additional questions, and consider raising the score if our explanations helped you resolve some concerns about our paper.

---

> > ### Author Rebuttal · Reviewer_KgXY · 2026-04-01
> >
> > The authors’ rebuttal addressed my concerns, so I increase my score from 4 to 5.

---

> > > ### Author Response · Authors · 2026-04-04
> > >
> > > ## Additional Results on Partially Correct Responses
> > >
> > > Here are our additional results on partially-correct responses. However, constructing such examples is nontrivial for graduate- or research-level tasks. We therefore build controlled hard negatives from verified correct human or strong-model solutions in the gold set: we truncate each solution at its 50% point, use Qwen3-4B-Thinking to complete the remaining reasoning, and retain only continuations whose final answer is incorrect. This produces examples that preserve substantial correct intermediate structure while remaining wrong overall. We then score these continuations with CBU and with LLM judges. Since CBU is natively in ([0,1]) while judge scores are on a 0 to 10 scale, we normalize judge scores by dividing by 10, so that 0 is the ideal value for both metrics on this benchmark. Under this construction, every evaluated continuation is incorrect, so the desired behavior is to assign scores as close to 0 as possible.
> > >
> > > **On this stress test, CBU is substantially more conservative than LLM judges. Across both GPT-OSS-20B and GPT-OSS-120B, CBU has median 0.0, whereas normalized LLM judges have medians 0.240 and 0.205; moreover, CBU assigns the lower score on 91.3% and 86.2% of paired examples, respectively.** All paired significance tests strongly reject equality. These results suggest that, CBU is much less prone than LLM judges to over-credit partially correct but ultimately incorrect solutions.
> > >
> > > All numbers below use normalized LLM judge scores, obtained by dividing raw 0 to 10 judge outputs by 10. Lower is better because every continuation is incorrect.
> > >
> > > ### Results on incorrect continuations
> > >
> > > | Backbone     | (n) |  CBU mean | LLM judge mean | CBU median | LLM judge median |
> > > | ------------ | --: | --------: | -------------: | ---------: | ---------------:|
> > > | GPT-OSS-20B  | 849 | **0.070** |          0.301 |  **0.000** |            0.240 |
> > > | GPT-OSS-120B | 835 | **0.097** |          0.276 |  **0.000** |            0.205 |
> > >
> > > ### Paired comparison
> > >
> > > | Backbone     | Mean paired diff (judge - CBU) | Median paired diff | CBU lower on | Wilcoxon (p) | Paired (t)-test (p) |
> > > | ------------ | -----------------------------: | -----------------: | -----------: | -----------: | ------------------: |
> > > | GPT-OSS-20B  |                          0.231 |              0.214 |        91.3% |   9.276e-102 |          2.591e-130 |
> > > | GPT-OSS-120B |                          0.179 |              0.195 |        86.2% |    1.131e-70 |           2.477e-68 |
> > >
> > > ---
> > >
> > > **Taken together, these results show that even on wrong responses containing substantial correct intermediate structure, CBU suppresses scores much more reliably than LLM judges.**
> > >
> > > Thank you again for your time to look at our paper. Please feel free to remind us if we have missed something.

---

### Official Review · Reviewer_PqFe · 2026-03-13

**Soundness:** 3
**Presentation:** 3
**Significance:** 3
**Originality:** 3
**Overall Recommendation:** 5
**Confidence:** 4

**Summary:**

This paper proposes a novel Consequence-Based Utility evaluation method, which scores an answer based on whether it is helpful for solving its neighborhood problems. The core idea is to assess answer quality through its downstream usefulness, rather than relying solely on direct preference estimation. Empirical results show that this method achieves significantly higher evaluation accuracy on challenging problems compared with traditional reward models and LLM-as-a-judge approaches. The paper also provides a thorough discussion of the method’s applicability, required conditions, and potential limitations.

**Compliance With Llm Reviewing Policy:**

Affirmed.

**Final Justification:**

These rebuttals addressed my concerns. I increase my score from 4 to 5.

**Key Questions For Authors:**

See weakness.

**Limitations:**

yes

**Strengths And Weaknesses:**

# Strengths

**Soundness**

1. The proposed Consequence-Based Utility evaluation method is novel and well motivated.

2. The paper shows substantial improvements over traditional reward models and LLM-as-a-judge on challenging problems.

3. The authors are careful in discussing the method’s assumptions, applicability, and limitations, which improves the overall credibility of the work.

**Presentation**

1. The paper is clearly written and easy to follow.


**Significance**

1. The paper offers a new perspective on evaluating difficult reasoning problems, where standard judges often struggle.

2. This could inspire future work on more reliable evaluation methods for hard tasks.

**Originality**

1. The core idea is highly original.


# Weaknesses

**Soundness**

1. CBU may capture not only correctness, but also how useful a candidate solution is for a particular solver. It is therefore unclear whether the method evaluates the intrinsic correctness of a solution or its solver-dependent in-context usefulness.

2. While the paper tests multiple backbones, it does not systematically analyze whether the ranking of the same candidate solutions is consistent across different solvers.

3. The comparison with reward models is not fully fair: the reward model appears to use a single deterministic score, while GenRM / LLM-Judge are repeated 64 times.

**Presentation**

No weakness.

**Significance**

1. The method may be difficult to scale in practice because it requires suitable neighborhood questions, which can be expensive to construct and may need careful difficulty control.

2. CBU also has noticeably higher inference cost than standard single-pass judges because it requires multiple rollouts.

3. The method seems sensitive to problem difficulty: on easier benchmarks, utility may lose discriminative power, which limits the range of settings where CBU is most useful.

**Originality**

No weakness.

---

> ### Author Rebuttal · Authors · 2026-03-31
>
> Dear Reviewer PqFe,
>
> Thank you for your time and effort in reviewing our paper. We have summarized our response to the 6 weaknesses you’ve mentioned as W1 to W6.
>
> ---
>
> ## [W1] CBU may also reward partial helpfulness, not only correctness
>
> To start with, Section 6 and Figure 4 partially covers this issue. In Section 6 we work with four PhD students to conduct a qualitative study on areas CBU outperform judges. Among 112 trajectories studied 77 of them (approx. 68.8%) included partially wrong reasoning. LLM-Judges have failed to identify such errors, while CBU correctly penalizes them. This shows that **CBU is more effective in identifying and penalizing partially-correct solutions compared to LLM-Judges.**
>
> Additionally, we are running additional experiments with LLM-generated partially wrong solutions and will be adding them shortly during the review period.
>
> ---
>
> ## [W2] The ranking consistency across different models
>
> Thanks for pointing this out. In the table below we computed pairwise correlations between CBU scores produced using different solver backbones on the same problem-candidate solutions pairs. The results show strong agreement across all model pairs, Pearson Corr. of 0.78 to 0.85, Spearman Corr. of 0.79 to 0.85, and Kendall Corr. of 0.67 to 0.75. These indicate that **CBU induces a largely stable ordering of candidate solutions across solvers rather than a ranking that is idiosyncratic to a particular backbone**.
>
> | Solver pair | Pearson | Spearman | Kendall |
> | --------------------------- | ------: | -------: | ------: |
> | GPT-OSS-120B vs. Qwen3-235B | 0.80 | 0.80 | 0.67 |
> | GPT-OSS-120B vs. Qwen3-30B | 0.78 | 0.79 | 0.67 |
> | Qwen3-235B vs. Qwen3-30B | 0.85 | 0.85 | 0.75 |
>
> ---
>
> ## [W3 & W5] Compute-Wise comparison with reward models
>
> First, we would like to clarify that the main comparison in Table 2 is budget-matched across the scalable evaluators. Specifically, GenRM and LLM-Judge are each repeated 64 times to match CBU’s rollout budget. Reward models are the only exception: we report them in their standard deterministic single-pass setting. This is because reward models do not natively support test-time scaling. **We view this as a limitation of plain reward models rather than an unfair advantage for CBU**, and Table 2 reflects this clearly: RMs are far weaker in HumanWin, with Q2.5-RM-72B at 1.63 and AceMath-72B-RM at 0.00, compared with 27.05 for Q3-235B-GenRM, 67.14 as an LLM-Judge, and 81.43 for CBU. Importantly, **Table 5 explicitly notes that CBU and LLM-Judge consume comparable numbers of tokens on average, so neither enjoys a systematic budget advantage; Figure 6 further shows that CBU converges at a similar rate overall, faster on some backbones, and that (n \ge 8) already keeps the mean normalized error below 0.05 across all tested models.** Thus, while CBU is more expensive than a literal single-pass judge, its comparison to GenRM and LLM-Judge is fair in inference budget, and in practice a small-rollout regime already captures most of the benefit.
>
> ---
>
> ## [W4] Trickiness to construct neighborhood questions
>
> In the paper, we consider three sources of neighborhood questions: expert-written questions (Section 4), LLM-generated variants, and questions derived from ancestor literature (Section 7.2). While expert-written questions are difficult to approach, the other two methods are reasonable. First, regarding the automated creation of questions by LLMs, several relevant work has already shown such. [1,2] We also retry ourselves in Table 4 and confirm that this is applicable in our setting. Second, we would like to highlight that it is very unlikely for one to encounter a scenario where either method is not possible. Conjectures with absolutely no previous work, are extremely rare in mathematics meaning our second method shell be usable in most cases.  Finally, **the most recommended area for CBU to be used is incremental-level math research**, the type of research which pushes an already existing domain further. **Our paper proves CBU to outperform traditional methods in such areas throughout Sections 4 to 5** by constructing our own dataset of faculty-authored research-level math and comparing it against LLM-Judges and GenRMs.
>
> [1] https://arxiv.org/abs/2505.12575
> [2] https://arxiv.org/abs/2602.24173v1
>
> ---
>
> ## [W6] Sensitive to problem difficulty
>
> We agree with that CBU is sensitive to problem difficulty and that, on easier benchmarks, it may lose discriminative power; this limitation is acknowledged in lines 400 to 405. However, our aim is not to replace existing evaluators on easier-level mathematics, but to improve evaluation on genuinely hard, research-level problems where standard evaluators are less reliable. In that intended regime, our results show that CBU remains effective. (Table 4)
>
> ---
>
> Please feel free to ask any additional questions, and consider raising the score if our explanations helped you resolve some concerns about our paper.

---

> > ### Author Rebuttal · Reviewer_PqFe · 2026-04-02
> >
> > Thank you for your response. My concerns have been addressed. I decide to maintain my score, because I believe it is a fair and positive score.

---

> > > ### Author Response · Authors · 2026-04-04
> > >
> > > ## Additional Results on Partially Correct Responses (W1)
> > >
> > > Here are our additional results on partially-correct responses. However, constructing such examples is nontrivial for graduate- or research-level tasks. We therefore build controlled hard negatives from verified correct human or strong-model solutions in the gold set: we truncate each solution at its 50% point, use Qwen3-4B-Thinking to complete the remaining reasoning, and retain only continuations whose final answer is incorrect. This produces examples that preserve substantial correct intermediate structure while remaining wrong overall. We then score these continuations with CBU and with LLM judges. Since CBU is natively in ([0,1]) while judge scores are on a 0 to 10 scale, we normalize judge scores by dividing by 10, so that 0 is the ideal value for both metrics on this benchmark. Under this construction, every evaluated continuation is incorrect, so the desired behavior is to assign scores as close to 0 as possible.
> > >
> > > **On this stress test, CBU is substantially more conservative than LLM judges. Across both GPT-OSS-20B and GPT-OSS-120B, CBU has median 0.0, whereas normalized LLM judges have medians 0.240 and 0.205; moreover, CBU assigns the lower score on 91.3% and 86.2% of paired examples, respectively.** All paired significance tests strongly reject equality. These results suggest that, CBU is much less prone than LLM judges to over-credit partially correct but ultimately incorrect solutions.
> > >
> > > All numbers below use normalized LLM judge scores, obtained by dividing raw 0 to 10 judge outputs by 10. Lower is better because every continuation is incorrect.
> > >
> > > ### Results on incorrect continuations
> > >
> > > | Backbone     | (n) |  CBU mean | LLM judge mean | CBU median | LLM judge median |
> > > | ------------ | --: | --------: | -------------: | ---------: | ---------------:|
> > > | GPT-OSS-20B  | 849 | **0.070** |          0.301 |  **0.000** |            0.240 |
> > > | GPT-OSS-120B | 835 | **0.097** |          0.276 |  **0.000** |            0.205 |
> > >
> > > ### Paired comparison
> > >
> > > | Backbone     | Mean paired diff (judge - CBU) | Median paired diff | CBU lower on | Wilcoxon (p) | Paired (t)-test (p) |
> > > | ------------ | -----------------------------: | -----------------: | -----------: | -----------: | ------------------: |
> > > | GPT-OSS-20B  |                          0.231 |              0.214 |        91.3% |   9.276e-102 |          2.591e-130 |
> > > | GPT-OSS-120B |                          0.179 |              0.195 |        86.2% |    1.131e-70 |           2.477e-68 |
> > >
> > > ---
> > >
> > > **Taken together, these results show that even on wrong responses containing substantial correct intermediate structure, CBU suppresses scores much more reliably than LLM judges.**
> > >
> > > Thank you again for your time to look at our paper. Please feel free to remind us if we have missed something. Also, if we have handled all your concerns, please consider raising the score.

---

### Decision · Program_Chairs · 2026-04-30

**Decision:**

Accept (spotlight)

**Comment:**

This paper addresses the verification bottleneck for research-level mathematics by proposing Consequence-Based Utility (CBU), an oracle-free evaluation method that scores candidate solutions based on their downstream utility as in-context exemplars for related, verifiable questions. The authors introduce EXPERTMATH, a high-quality dataset of expert-annotated problems, and demonstrate that CBU consistently outperforms existing oracle-free baselines (reward models, generative reward models, and LLM judges) in ranking quality, achieving substantial improvements in Acc@1 and AUC while maintaining robust correct–wrong separation. All reviewers unanimously recommend acceptance, praising the methodological novelty, rigorous empirical validation, practical implementation guidelines, and valuable dataset contribution. The work offers a principled and scalable approach to LLM solution verification, strongly aligning with ICML’s focus on reliable evaluation methods for advanced reasoning, and is recommended for acceptance.